# Structural mechanism of DDX39B regulation by human TREX-2 and a related complex in mRNP remodeling

Bradley P. Clarke [1], Shengyan Gao[2], Menghan Mei [1], Dongqi Xie [2], Alexia E. Angelos[1], Ashley Vazhavilla[2], Pate S. Hill[1], Tolga Cagatay [2], Kimberly Batten[2], Jerry W. Shay [2], Yihu Xie [1] ✉, Beatriz M. A. Fontoura [2] ✉ & Yi Ren [1,3] ✉

Nuclear export of mRNAs in the form of messenger ribonucleoprotein particles (mRNPs) is an obligatory step for eukaryotic gene expression. The DEAD-box ATPase DDX39B (also known as UAP56) is a multifunctional regulator of nuclear mRNPs. How DDX39B mediates mRNP assembly and export in a controlled manner remains elusive. Here, we identify a novel complex TREX-2.1 localized in the nucleus that facilitates the release of DDX39B from the mRNP. TREX-2.1 is composed of three subunits, LENG8, PCID2, and DSS1, and shares the latter two subunits with the nuclear pore complex-associated TREX-2 complex. Cryo-EM structures of TREX-2.1/DDX39B and TREX-2/DDX39B identify a conserved trigger loop in the LENG8 and GANP subunit of the respective TREX-2.1 and TREX-2 complex that is critical for DDX39B regulation. RNA sequencing from LENG8 knockdown cells shows that LENG8 influences the nucleocytoplasmic ratio of a subset of mRNAs with high GC content. Together, our findings lead to a mechanistic understanding of the functional cycle of DDX39B and its regulation by TREX-2 and TREX-2.1 in mRNP processing.

In eukaryotes, mRNAs are packaged with proteins into messenger ribonucleoprotein particles (mRNPs). The processes governing the assembly and remodeling of mRNPs are fundamentally important for the entire life cycle of mRNA. Different binding proteins are required to regulate each stage, from transcription, processing, and nuclear export to translation[1–4]. Nuclear mRNP assembly occurs co-transcriptionally and involves many RNA-binding proteins[5–7]. Following proper processing of the mRNA and acquisition/removal of specific proteins, export-competent mRNPs travel from the nucleus to the cytoplasm through the nuclear pore complex (NPC)[8–15].

In the nucleus, newly synthesized mRNA is bound by the nuclear cap-binding complex (CBC) at the 5′-cap shortly after the start of transcription[16–19]. In mammalian cells, the RNA-binding protein ALYREF

directly recognizes the CBC to recruit a crucial mRNA export factor, the TREX (TRanscription-EXport) complex[20,21]. TREX is composed of the THO subcomplex, the DEAD-box ATPase DDX39B (also known as UAP56; yeast Sub2), and ALYREF (yeast Yra1)[22–26]. TREX and its associated factors play key roles in nuclear mRNP packaging. In both humans and yeast, THO facilitates the recruitment of DDX39B or Sub2 to actively transcribed genes[23,27,28]. The functional states of DDX39B/Sub2 are tightly coupled to its ATPase cycle. THO directly binds to DDX39B/Sub2 and positions its two RecA domains in a half-open conformation, as first revealed by a crystal structure of the yeast THO/Sub2 complex[24]. Later, multiple cryo-EM structures of THO and DDX39B/Sub2 from both human and yeast proteins confirmed the conservation of this mechanism[25,29–32]. When engaged with the mRNP,

[1]Department of Biochemistry, Vanderbilt University School of Medicine, Nashville, TN, USA. [2]Department of Cell Biology, University of Texas Southwestern Medical Center, Dallas, TX, USA. [3]Center for Structural Biology, Vanderbilt University, Nashville, TN, USA. ✉e-mail: yihu.xie@vanderbilt.edu; Beatriz.Fontoura@UTSouthwestern.edu; yi.ren@vanderbilt.edu

both human DDX39B and yeast Sub2 adopt a closed conformation, as revealed by crystal structures from the DDX39B/Tho1/RNA and Sub2/Yra1/RNA complexes[24,26]. ALYREF/Yra1 and some SR proteins can act as adaptors that bridge the export receptor heterodimer NXF1-NXT1 (yeast Mex67-Mtr2) to the mRNP[33–37]. NXF1-NXT1/Mex67-Mtr2 can directly interact with the FG domains of nucleoporins, and its presence as an mRNP component is required for translocation through the NPC[38,39].

The network of interactions surrounding DDX39B and the THO complex mediates the coupling of mRNP packaging with transcription, splicing, and 3′-end processing[40–46]. In yeast, phosphorylation on the C-terminal domain (CTD) of RNA Pol II regulates its association with the THO complex[47,48]. In humans, the phosphorylated CTD of RNA Pol II was also shown to be important for co-transcriptional mRNP biogenesis[49–51]. The TREX complex interacts with not only mRNP assembly factors, but also some factors involved in mRNP quality control, including the yeast SR protein Gbp2, the human polyA binding protein ZC3H14 (yeast Nab2), and human ZC3H18[25,43,48,52–57]. The majority of human genes have introns, and splicing has been shown to promote mRNP packaging and export[23,28,58,59]. The mRNP biogenesis factors are extensively connected and likely contribute to the crosstalk between mRNP processing and export. For instance, the spliceosome-associated factor U2AF2 (yeast Mud2) interacts with DDX39B/Sub2 and THO to regulate splicing and mRNP assembly[60–64]. ZC3H14/Nab2 was shown to interact with both U2AF2 and the THO complex to couple 3′-end processing with splicing and mRNP export[43,65,66]. As central players in mRNP metabolism, DDX39B and some other TREX components are required for the release of spliced transcripts from the nuclear speckles for export[67–69]. To date, how these multiple functions of DDX39B during mRNP packaging are coordinated remains poorly understood.

The final steps of mRNP nuclear export are initiated by the mRNP docking at the nuclear basket of the NPC[9,70–74]. The nuclear basket-associated TREX-2 complex directly interacts with NXF1-NXT1, which facilitates mRNP entry to the NPC channel[75–78]. Localization of human TREX-2 at the NPC requires the nucleoporins Nup153 and TPR[70,76,79,80]. Interestingly, both human TPR and the yeast ortholog Mlp1 are also shown to function in mRNP quality control[80,81]. In particular, a direct interaction between Mlp1 and Nab2 is involved[82–85]. A functional link between Nab2 and TREX-2 further supports the integrated nature of these processes[86]. Studies on the Balbiani ring mRNPs suggest that DDX39B accompanies the mRNP to the nuclear pore, but is released prior to export through the NPC[87]. For cellular mRNPs, how the final remodeling events are coordinated at the NPC, including DDX39B removal and NXF1-NXT1 loading, remains elusive. Overall, the coordinated production of nuclear mRNPs involves many levels of regulation mediated by multi-functional factors, including the CBC, TREX, TREX-2, DDX39B/Sub2, NXF1-NXT1/Mex67-Mtr2, and ZC3H14/Nab2. A better understanding of the interactions between these factors has the potential to elucidate the underlying sequence of molecular events required for mRNP nuclear export.

Here, we successfully reconstituted an alternative TREX-2 complex composed of LENG8, PCID2, and DSS1, which we term the TREX-2.1 complex. We show that the TREX-2.1 complex mainly localizes throughout the nucleus and can function together with DDX39B to remodel mRNPs. Cryo-EM structures of TREX-2 and TREX-2.1 in complex with DDX39B allow us to identify a conserved trigger loop from their respective GANP and LENG8 subunits as a molecular determinant for the mRNP remodeling activity. We found that TREX-2.1 influences the nucleocytoplasmic distribution of a subset of mRNAs with high GC content. Furthermore, Influenza A Virus (IAV) NP protein targets the mRNP remodeling steps involving DDX39B with TREX-2 or TREX-2.1. Together, our studies provide new insights into the structural and functional roles of TREX-2 and a novel TREX-2.1 complex in mRNP processing via regulation of DDX39B.

## Results

### LENG8 forms an alternative TREX-2 complex with PCID2/DSS1

TREX-2 is composed of five subunits[88] (Fig. 1a). The largest subunit, GANP (yeast Sac3), serves as a scaffold that brings together the other four subunits. The N-terminal region of GANP/Sac3 contains FG repeats and interacts with the mRNA export receptor NXF1-NXT1[75–77]. The middle region of GANP/Sac3 adopts a PCI fold and interacts with PCID2 (yeast Thp1) and DSS1 (yeast Sem1) to form an M sub-complex[89–91]. The CID domain of GANP/Sac3 forms a complex with ENY2 (yeast Sus1) and Centrins (yeast Cdc31)[92,93]. In yeast, Nup1 directly binds to the CID domain of TREX-2, which is responsible for its NPC targeting[94].

The GeneCards database suggests that GANP has a paralog, LENG8[95]. The most conserved region between GANP and LENG8 is the PCI fold, though the homology is modest (~20% identity) (Fig. 1a). Unlike the well-studied GANP and the TREX-2 complex in mRNA export, the function of LENG8 is largely unknown. Through affinity purification, co-immunoprecipitation, or yeast two-hybrid approaches, LENG8 was shown to associate with various mRNP biogenesis factors, such as the THO complex, ZC3H11A, U2AF1, ZC3H18, ZFC3H1, ZCCHC8, EXOSC3, DSS1, and PCID2[57,96–102]. These studies suggest that the function of LENG8 is linked to mRNP processing. However, it is not known whether the reported interactions are direct or mediated by nucleic acids and/or other factors.

We tested some of the above putative binding partners of LENG8 with purified recombinant proteins. We found that GST-LENG8 cannot pull down the THO complex (Fig. 1b), suggesting that though this interaction has been found in cells, the interaction is likely mediated by other factors. Consistently, RNase treatment greatly weakened the interaction between LENG8 and the THO complex components in co-IP experiments[99]. In contrast to the THO complex, we found that GST-LENG8 can pull down the PCID2/DSS1 complex (Fig. 1b, Supplementary Fig. 1a). Next, we successfully reconstituted a recombinant LENG8/PCID2/DSS1 complex through co-expression in *E. coli*. We further purified the complex to homogeneity as shown by size exclusion chromatography (Fig. 1c). Hereafter, this TREX-2-related hetero-trimeric LENG8/PCID2/DSS1 complex is referred to as the TREX-2.1 complex (Fig. 1a).

LENG8 (800 residues) is significantly shorter than GANP (1980 residues), missing both the N-terminal NXF1-NXT1 binding domain and the C-terminal NPC targeting domain of GANP (Fig. 1a). We assessed the subcellular localization of endogenous LENG8 by immuno-fluorescence (Fig. 1d) and found that LENG8 is mostly localized inside the nucleus. This observation is consistent with the lack of an NPC targeting domain in LENG8. The mechanism of action of this novel TREX-2.1 complex is largely unknown. To better understand the dif-ferential roles of TREX-2 and the related TREX-2.1 complex in nuclear mRNP maturation and export, we carried out functional studies.

### TREX-2.1 complex remodels mRNPs through direct interaction with DDX39B

The Sac3 subunit of the yeast TREX-2 complex (ortholog of human GANP) was shown to associate with Sub2 (ortholog of human DDX39B) isolated from yeast cells[22,75]. We tested the possible direct interaction between human DDX39B and TREX-2.1 with purified recombinant proteins. We found that GST-DDX39B can pull down TREX-2.1 effi-ciently (Fig. 2a). Additionally, we performed immunoprecipitation using 293T cells transfected with Flag-LENG8 and HA-PCID2. LENG8 was able to co-immunoprecipitate with PCID2 and DDX39B (Fig. 2b). Together, our studies show that TREX-2.1 interacts with DDX39B.

We next asked how TREX-2.1 affects the activity of DDX39B. To this end, we pre-assembled a DDX39B/RNA complex in the presence of an ATP analog, ATP-γ-S. We found that TREX-2.1 can displace DDX39B from the DDX39B/RNA complex (Fig. 2c). This activity requires the integrity of the TREX-2.1 complex, as LENG8 alone or the PCID2/DSS1

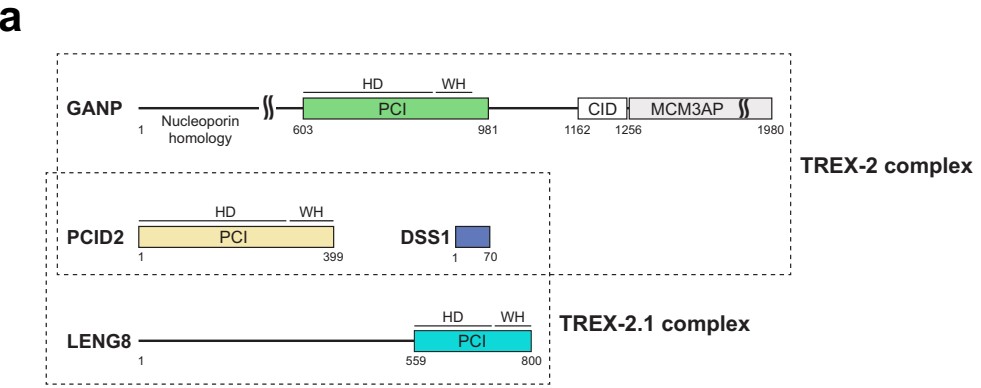

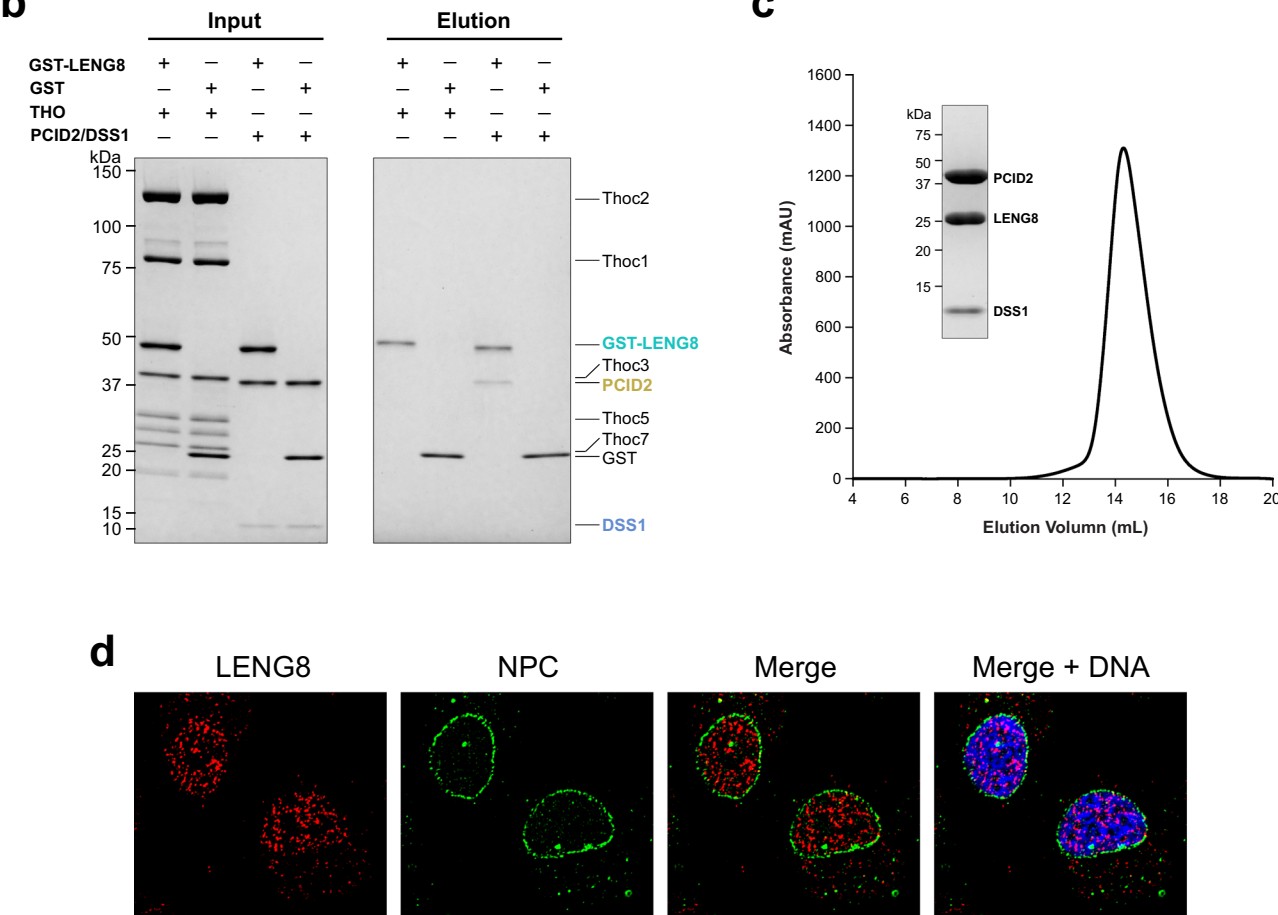

**Fig. 1 | Human TREX-2 and related TREX-2.1 complexes. a** Schematic representations of the TREX-2 complex and the novel TREX-2.1 complex. TREX-2 and TREX-2.1 each contain a unique scaffold subunit, GANP and LENG8, respectively. GANP can be divided into three regions: the N-terminal region containing FG nucleoporin-like motifs, the middle region featuring a PCI fold formed by a helical domain (HD) and a winged-helix domain (WH), and the C-terminal region composed of CID and MCM3AP domains. The CID domain forms a complex with Centrins and two copies of ENY2 (subunits of TREX-2, omitted in the graph for clarity). LENG8 contains a predicted unstructured N-terminal region and a C-terminal PCI fold, which also forms a complex with PCID2 and DSS1, as shown in (**b**, **c**). **b** LENG8 forms a novel TREX-2.1 complex with PCID2 and DSS1. In vitro GST pull-down was performed with purified GST-LENG8 and human THO complex or the PCID2/DSS1 complex. GST-LENG8 binds to PCID2/DSS1, but not the THO complex. **c** Purification of the TREX-2.1 complex containing LENG8, PCID2, and DSS1. Purified TREX-2.1 complex was analyzed by size exclusion chromatography using a Superdex 200 column. Coomassie-stained SDS-PAGE of the peak fraction is shown. **d** LENG8 localization in A549 cells. Cells were subjected to immunofluorescence microscopy with anti-LENG8 (red) and anti-NPC (green) as indicated. Scale bar, 10 μm. Experiments in (**b**–**d**) have been repeated three times independently with similar results. Source Data are provided as a Source Data file.

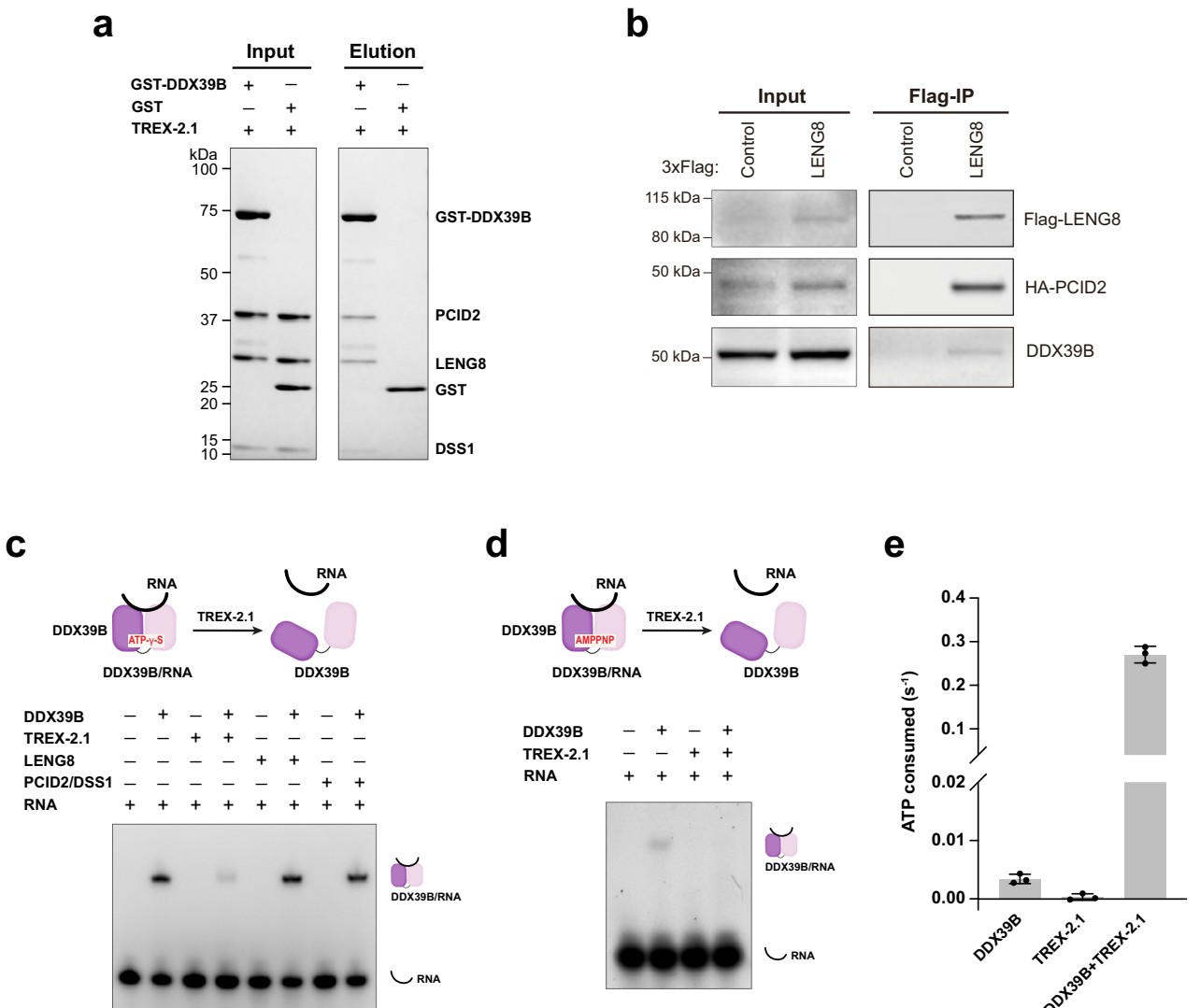

**Fig. 2 | TREX-2.1 regulates the activity of DDX39B. a** TREX-2.1 directly interacts with DDX39B. In vitro GST pull-down was performed with purified GST-DDX39B and TREX-2.1. **b** LENG8 co-immunoprecipitates with PCID2 and DDX39B. 293T cells transfected with a FLAG-LENG8 plasmid or an empty Flag plasmid control and with an HA-PCID2 plasmid were subjected to immunoprecipitation with anti-Flag antibody. **c** TREX-2.1 displaces DDX39B from a pre-assembled DDX39B/RNA/ATP-γ-S complex. The integrity of the TREX-2.1 complex is required for DDX39B displacement from the DDX39B/RNA/ATP-γ-S complex. **d** TREX-2.1 displaces DDX39B from a pre-assembled DDX39B/RNA/AMPPNP complex. Experiments in (**a**–**d**) have been repeated three times independently with similar results. **e** TREX-2.1 stimulates the ATPase activity of DDX39B. Each column represents the mean of 3 independent replicates. Error bars represent standard deviation. Source Data are provided as a Source Data file.

subcomplex alone could not unload DDX39B from the DDX39B/RNA complex (Fig. 2c). Furthermore, TREX-2.1 can displace DDX39B from the DDX39B/RNA complex pre-assembled in the presence of a non-hydrolyzable analog, AMPPNP, suggesting that the mRNP remodeling activity of TREX-2.1 on DDX39B does not require ATP hydrolysis (Fig. 2d).

We further tested the effect of TREX-2.1 on the ATPase activity of DDX39B and found that TREX-2.1 significantly enhanced the ATPase activity of DDX39B, increasing it by about 80-fold (Fig. 2e). Several DEAD-box ATPase family members, including Sub2, can form long-lived complexes on RNA[103]. DDX39B is required for mRNP export at both nuclear speckles and the NPC. Release of DDX39B from nuclear mRNPs is likely an important step subject to regulation. A well-studied DEAD-box ATPase involved in mRNP metabolism is the eIF4A3 from the exon junction complex (EJC). The EJC is deposited on spliced mRNAs in the nucleus. eIF4A3 hydrolyzes ATP inside the EJC but remains locked on RNA until the EJC disassembly factor PYM triggers dissociation of eIF4A3 from the mRNAs in the cytoplasm[104–108]. We

envision that TREX-2 and TREX-2.1 both act as disassembly factors for DDX39B-containing mRNPs to promote the release of DDX39B, but in different functional contexts based on their different localization. To understand the molecular basis for the observed remodeling activities, we carried out structural studies on the complexes of DDX39B with TREX-2 or TREX-2.1.

### Cryo-EM structure of the TREX-2^M/DDX39B complex

We first sought to determine the structure of the TREX-2/DDX39B complex. Full-length DDX39B and a TREX-2 complex containing the middle region of GANP (residues 358–1000), PCID2, and DSS1, hereafter referred to as TREX-2^M, were used for cryo-EM studies. The interaction between TREX-2^M and DDX39B is dynamic, and thus we employed a photoactivatable unnatural amino acid, *p*Bpa, to stabilize the TREX-2^M/DDX39B complex through photo-crosslinking. Site-specific labeling with *p*Bpa was performed using amber codon suppression[109] (Supplementary Fig. 2a). We found that DDX39B with *p*Bpa incorporated at residue 108 (DDX39B^*p*Bpa108) on its RecA1 domain

yielded robust crosslinking to TREX-2[M] upon UV treatment (Fig. 3a). The resulting photo-crosslinked TREX-2[M]/DDX39B complex was subjected to cryo-EM studies. ADP was included in the cryo-EM sample to mimic the post-hydrolysis state of DDX39B.

Sorting of the cryo-EM data yielded a map of the TREX-2[M]/DDX39B complex with an overall resolution of 3.25 Å (Fig. 3b, Supplementary Figs. 1–3, and Table 1). DDX39B contains an N-terminal motif (NTM) and two RecA-like domains (RecA1 and RecA2) that form the ATPase core. The cryo-EM map reveals that both the NTM and the two RecA domains of DDX39B make direct contact with the TREX-2[M] complex. Of note, the RecA1 domain of DDX39B binds to the concave face of the V-shaped TREX-2[M] complex, contacting both GANP and PCID2. The RecA2 domain of DDX39B flexibly binds to the GANP subunit. It is evident that the two RecA domains of DDX39B adopt an open conformation that is not compatible with the closed conformation observed when DDX39B is in complex with RNA[26].

We obtained a set of particles featuring the best densities for the RecA1 domain of DDX39B, which yielded a cryo-EM map of the TREX-2[M]/DDX39B[NTM+RecA1] complex with an overall resolution of 2.79 Å (Fig. 3c, d, Supplementary Figs. 2–4, and Table 1). The resulting TREX-2[M]/DDX39B[NTM+RecA1] structure shows that DDX39B is in its post-hydrolysis state, bound to ADP and features extensive interactions with TREX-2[M] involving both the NTM and RecA1 (Fig. 3e–h). The N-terminal segment of the NTM (NTM-N, residues 9–16) (Fig. 3e) contains several acidic residues (E9, D12, E14, D15, and D16) and binds to the basic groove formed by the WH domains of GANP and PCID2. The NTM-N interface also features hydrophobic interactions involving Y13, which is buried at the interface with PCID2. This NTM-N is connected via an intrinsically flexible region to the C-terminal segment of the NTM (NTM-C, residues 38–45) (Fig. 3f) that lies on the surface of GANP. The NTM-C connects to the RecA1 domain of DDX39B, which is cradled by GANP and PCID2 with a buried surface area of 2202 Å$^2$ (Fig. 3g). The RecA1 interface with GANP involves the ADP binding site of RecA1 and several loops at the N-terminus of the GANP PCI fold (Fig. 3g). An extended loop of GANP (residues 674–688), hereafter referred to as the trigger loop, directly contacts ADP (Fig. 3g. left, Fig. 3h). The R678 residue of the trigger loop stacks against the adenine moiety of ADP via cation–π interaction. Additionally, the Y676 residue is positioned in a configuration for cation–π interaction with R728 of GANP, which also mediates a salt bridge with D66 of DDX39B. The main-chain carbonyl group of S680 forms a hydrogen bond with R135 of DDX39B. Furthermore, the E685 residue forms a salt bridge with K138 of DDX39B. The RecA1 interface with PCID2 involves extensive interactions (Fig. 3g, right). L51 and M83 of RecA1 make key hydrophobic interactions with PCID2. Q105 and Q106 of RecA1 form salt bridges with E143 and R150 of PCID2, respectively. The structure also reveals that the incorporated pBpa108 on the RecA1 domain DDX39B is crosslinked to the R93 residue of PCID2 (Supplementary Fig. 4b, right).

To highlight the differences in how DDX39B's RecA domains are oriented and the impact that TREX-2[M] binding has on their conformational state, we compare the TREX-2[M]/DDX39B[NTM+RecA1] structure with the DDX39B/RNA complex structure by aligning the RecA1 domain of DDX39B (Fig. 4a). For the RNA-bound DDX39B, both RecA domains contribute to the highly coordinated interactions with the nucleotide and RNA, which are located on the opposite side of DDX39B. Intriguingly, R678 from the trigger loop of GANP would break key interactions between ADP and the F381 residue from DDX39B–RecA2 of the DDX39B/RNA complex (Fig. 4a–d). It is well known that RNA binding of DEAD-box ATPases is tightly coupled with the coordinated interactions with the nucleotides[110–112]. The decoupling of nucleotide binding of DDX39B would lead to RNA release. Our structure readily suggests an explanation for the observed remodeling and ATPase-stimulating activities of TREX-2.1 on the DDX39B/RNA

complex (Fig. 2c–e). The intermediate state trapped in the structure of TREX-2[M]/DDX39B[NTM+RecA1] indicates that TREX-2[M] can disengage the RecA2 domain of DDX39B from the nucleotide and thus promote the release of the ATPase from RNA.

The role of the GANP trigger loop was assessed with an ATPase assay (Fig. 4e). Indeed, a TREX-2[M] complex with a truncated trigger loop (TREX-2[M]-mut, [678]RSSADQEEP[686] replaced with GGGG) showed about 40-fold lower activation of DDX39B. We also tested DDX39B activation with the GANP subunit of TREX-2[M], which contains the trigger loop (Fig. 4e). GANP alone failed to activate DDX39B, suggesting that interactions from the other subunits of TREX-2[M], PCID2/DSS1, are important to support the activity. The PCID2/DSS1 subcomplex also failed to activate DDX39B (Fig. 4e). Taken together, the extensive interactions between all subunits of TREX-2[M] and DDX39B are required to configure the trigger loop in the right conformation and position for DDX39B regulation. Nucleotide release is a rate-limiting step for DEAD-box ATPases[113–117]. Our results suggest that TREX-2[M] directly engages with the nucleotide and likely destabilizes the interaction between the nucleotide and DDX39B, which would promote the recycling and ATPase activity of DDX39B.

Regarding the related TREX-2.1 complex, the LENG8 subunit is significantly shorter than the corresponding GANP subunit of the TREX-2 complex. To further understand the molecular mechanism underlying DDX39B regulation by TREX-2 and related complexes, we carried out structural studies of human TREX-2.1 in complex with DDX39B.

## Cryo-EM structure of the TREX-2.1/DDX39B complex

We used a similar photo-crosslinking approach to obtain a TREX-2.1/DDX39B complex for cryo-EM studies (Supplementary Fig. 5a). Data processing yielded two major classes: TREX-2.1/DDX39B[NTM-N] and TREX-2.1/DDX39B (Supplementary Fig. 5b).

A 3.08 Å cryo-EM map of TREX-2.1/DDX39B[NTM-N] complex reveals an asymmetric V-shaped architecture of TREX-2.1 and that the NTM-N of DDX39B binds to the LENG8/PCID2 interface (Fig. 5a, b, Supplementary Fig. 6, Table 1). The LENG8 subunit forms the shorter leg of the V-shape, and the PCID2/DSS1 subcomplex forms the longer leg, with a buried surface area of 1362 Å$^2$ between them. The WH domains within the PCI fold of LENG8 and PCID2 mediate the primary interactions for the TREX-2.1 complex assembly (Fig. 5c, top). Key LENG8 residues at the interface involve Y711, A743, F748, and Y756. Additionally, the HD domains within the PCI fold also contribute to the LENG8-PCID2 interactions (Fig. 5c, bottom). In particular, R703 of LENG8 forms a salt bridge with E315 of PCID2; I679, T680, L683, and A684 of LENG8 mediate hydrophobic interactions with PCID2. Comparison of the TREX-2[M] and TREX-2.1 structures shows that they resemble each other in terms of the overall architecture, but LENG8 features two major differences from GANP (Fig. 5d, Supplementary Fig. 7). First, LENG8 is significantly shorter than GANP, with the lack of a GANP N-terminal region (Fig. 5d, GANP[N]) that is involved in DDX39B–RecA1 interaction (Fig. 3d, g). Second, while the LENG8 protein ends at the PCI fold at the C-terminus, GANP contains an extended C-terminal region (Fig. 5d, GANP[C], residues 937–981) that extends from the GANP-PCID2 interface to the DDX39B–RecA1 binding site on GANP.

A cryo-EM map of TREX-2.1/DDX39B, refined to an overall resolution of 3.28 Å, reveals that the RecA1 domain of DDX39B binds to the concave face of the V-shaped TREX-2.1 and that the RecA2 domain binds to the tip of the LENG8 leg with significant flexibility (Fig. 6a, Supplementary Fig. 8, Table 1). Comparison of TREX-2.1/DDX39B and TREX-2[M]/DDX39B indicates different modes of binding to DDX39B–RecA2 (Fig. 6b), likely due to the different architectures between LENG8 and GANP. We also obtained a set of particles that feature the best densities for the RecA1 domain of DDX39B, which

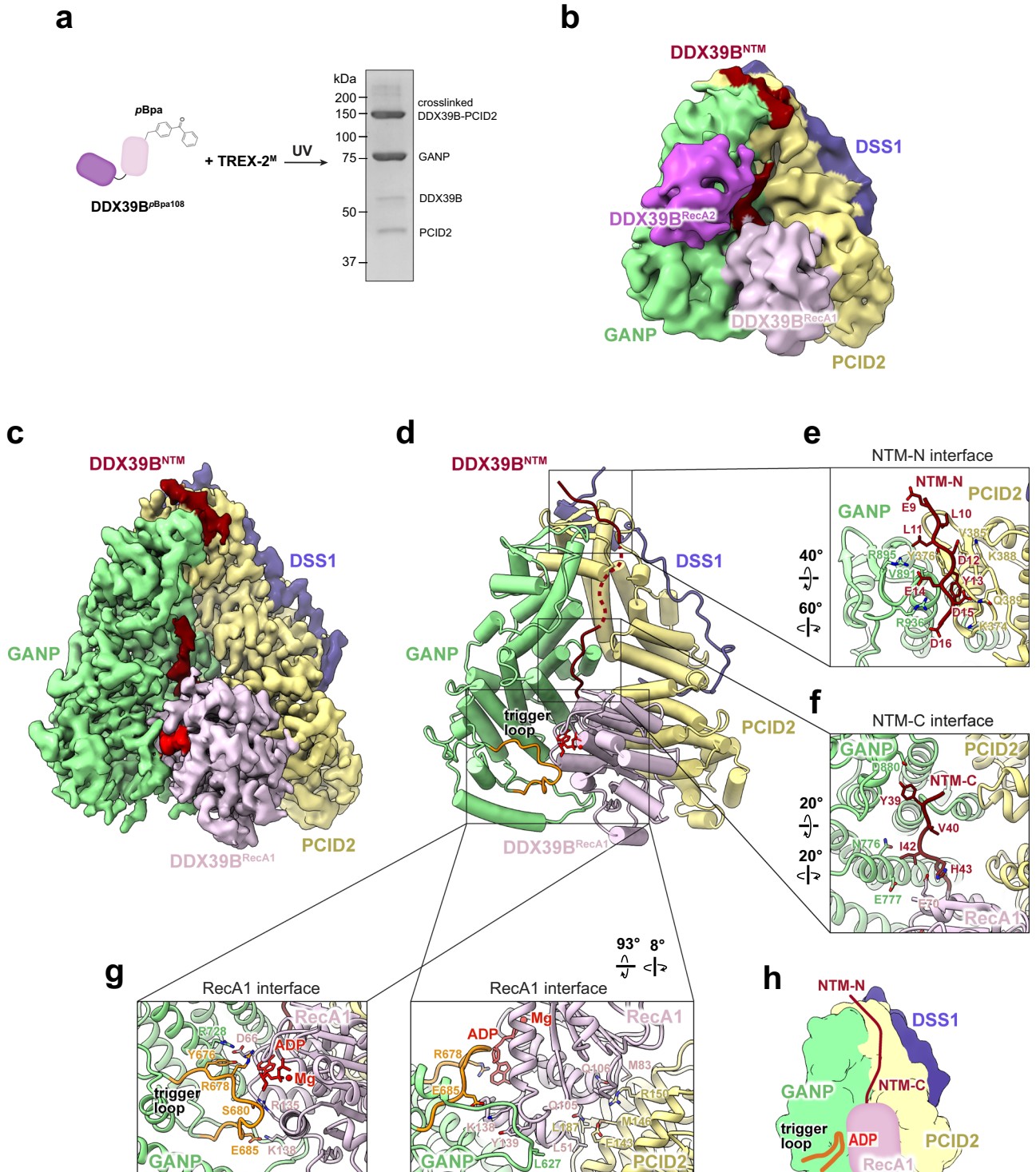

**Fig. 3 | Cryo-EM structure of the human TREX-2$^M$/DDX39B complex with ADP.**
**a** Schematics for generating the photo-crosslinked TREX-2$^M$/DDX39B complex and Coomassie-stained SDS-PAGE of the cryo-EM sample. This experiment has been repeated three times independently with similar results. **b** Cryo-EM map of the TREX-2$^M$/DDX39B complex determined at 3.25 Å resolution (EMD-46981), low-pass filtered at 6 Å resolution. The GANP, PCID2, and DSS1 subunits of TREX-2$^M$ are colored in green, yellow, and blue, respectively. The NTM, RecA1, and RecA2 domains of DDX39B are colored in red, pink, and purple, respectively. **c** Cryo-EM map of the TREX-2$^M$/DDX39B$^{NTM+RecA1}$ complex determined at 2.79 Å resolution (EMD-46982). **d** Structure of the TREX-2$^M$/DDX39B$^{NTM+RecA1}$ complex (PDB 9DLP). DDX39B is observed in a post-hydrolysis state bound to ADP. **e–g** TREX-2$^M$ interfaces with NTM-N (**e**), NTM-C (**f**), and RecA1 (**g**, arrowhead indicates the trigger loop) of DDX39B. **h** Schematics of the interactions between TREX-2$^M$ and DDX39B$^{NTM+RecA1}$. Source Data are provided as a Source Data file.

yielded a cryo-EM map of the TREX-2.1/DDX39B$^{NTM+RecA1}$ complex with an overall resolution of 2.97 Å (Fig. 6c, d, Supplementary Figs. 8 and 9, Table 1). While the interaction interfaces between DDX39B and PCID2 are the same for TREX-2$^M$ and TREX-2.1, TREX-2.1 features unique interactions at the LENG8 interfaces with DDX39B (Fig. 6e–g). For example, the NTM-C of DDX39B in the TREX-2.1/DDX39B$^{NTM+RecA1}$ complex is sandwiched between LENG8 and the RecA1 domain of DDX39B (Fig. 6d, f), while the NTM-C features a more extended

**Table 1 | Cryo-EM data collection, processing, and refinement statistics of the TREX-2$^M$/DDX39B and TREX-2.1/DDX39B complexes**

| | TREX-2$^M$/DDX39B$^{NTM+RecA1}$ (EMDB EMD-46982) (PDB 9DLP) | TREX-2$^M$/DDX39B (EMDB EMD-46981) | TREX-2.1/DDX39B$^{NTM+RecA1}$ (EMDB EMD-46985) (PDB 9DLV) | TREX-2.1/DDX39B$^{NTM-N}$ (EMDB EMD-46983) (PDB 9DLR) | TREX-2.1/DDX39B (EMDB EMD-47126) |
|---|---|---|---|---|---|
| **Data collection and processing** | | | | | |
| Microscope | Krios G4 | | Krios G4 | | |
| Voltage (kV) | 300 | | 300 | | |
| Camera | Gatan K3 | | Gatan K3 | | |
| Electron exposure (e$^-$/Å$^2$) | 52.9 | | 56.1 | | |
| Defocus range (μm) | −0.8 to −2.0 | | −0.8 to −2.0 | | |
| Pixel size (Å) | 0.820 | | 0.820 | | |
| Initial particle images (no.) | 22,639,463 | | 19,092,422 | | |
| Final particle images (no.) | 720,007 | 199,374 | 557,019 | 569,284 | 191,817 |
| Symmetry imposed | C1 | C1 | C1 | C1 | C1 |
| Box size (pixels) | 288 | 288 | 288 | 288 | 288 |
| Map resolution (masked, Å) | 2.79 | 3.25 | 2.97 | 3.08 | 3.28 |
| FSC threshold | 0.143 | 0.143 | 0.143 | 0.143 | 0.143 |
| **Refinement** | | | | | |
| Initial model used | AlphaFold model and PDB 8ENK | | AlphaFold model and PDB 8ENK | AlphaFold model | |
| Model resolution (Å) | 2.9 | | 3.2 | 3.3 | |
| FSC threshold | 0.5 | | 0.5 | 0.5 | |
| Map sharpening B factor (Å$^2$) | −50 | | −60 | −60 | |
| Model-to-map CC (mask) | 0.85 | | 0.82 | 0.80 | |
| Model composition | | | | | |
| Protein residues | 1049 | | 896 | 676 | |
| Ligand | 2 | | | | |
| B factors (Å$^2$) | | | | | |
| Protein | 105.6 | | 89.5 | 118.8 | |
| Ligand | 97.7 | | | | |
| R.m.s. deviations | | | | | |
| Bond lengths (Å) | 0.002 | | 0.002 | 0.003 | |
| Bond angles (°) | 0.429 | | 0.431 | 0.427 | |
| Validation | | | | | |
| MolProbity score | 1.28 | | 1.68 | 1.46 | |
| Clashscore | 5.00 | | 8.84 | 6.32 | |
| Poor rotamers (%) | 0.7 | | 0 | 0 | |
| Ramachandran plot | | | | | |
| Favored (%) | 98.3 | | 96.7 | 97.5 | |
| Allowed (%) | 1.7 | | 3.3 | 2.5 | |
| Disallowed (%) | 0 | | 0 | 0 | |

conformation in the TREX-2$^M$/DDX39B complex (Fig. 3d, f). At the RecA1 interface, a key interaction is mediated by Y624 of LENG8 and D66 of DDX39B (Fig. 6g). As LENG8 is significantly shorter at the N-terminal region (Fig. 5d), it lacks some of the RecA1 domain interactions found in GANP.

Comparison of the TREX-2.1/DDX39B$^{NTM-N}$ and the TREX-2.1/DDX39B$^{NTM+RecA1}$ structures reveals the structural flexibility of LENG8 (Supplementary Fig. 9d). Upon binding to DDX39B, the N-terminal helical domain of LENG8, located at the interface of NTM-C and RecA1, swings towards the RecA1 domain of DDX39B. Additionally, the N-terminal helical domain of PCID2 also features modest movement to accommodate the RecA1 interaction.

Structural and sequence alignments between LENG8 and GANP suggest that LENG8 contains an equivalent trigger loop (residues 559–570): a conserved YxR motif at the N-terminus as well as enriched proline residues at the C-terminus (Fig. 7a, b, Supplementary Fig. 7). Structural overlay of TREX-2.1/DDX39B and TREX-2$^M$/DDX39B shows that the trigger loop in LENG8 is positioned in proximity to the nucleotide binding site of DDX39B (Fig. 7b). The LENG8 trigger loop in the TREX-2.1/DDX39B complex does not show significant electron density, suggesting its dynamic nature. To assess the role of the trigger loop in TREX-2.1 function, we generated a mutant TREX-2.1 complex (TREX-2.1-mut), in which $^{563}$RLTCAP$^{568}$ in LENG8 is replaced by a GGGG linker. Compared to the wild-type

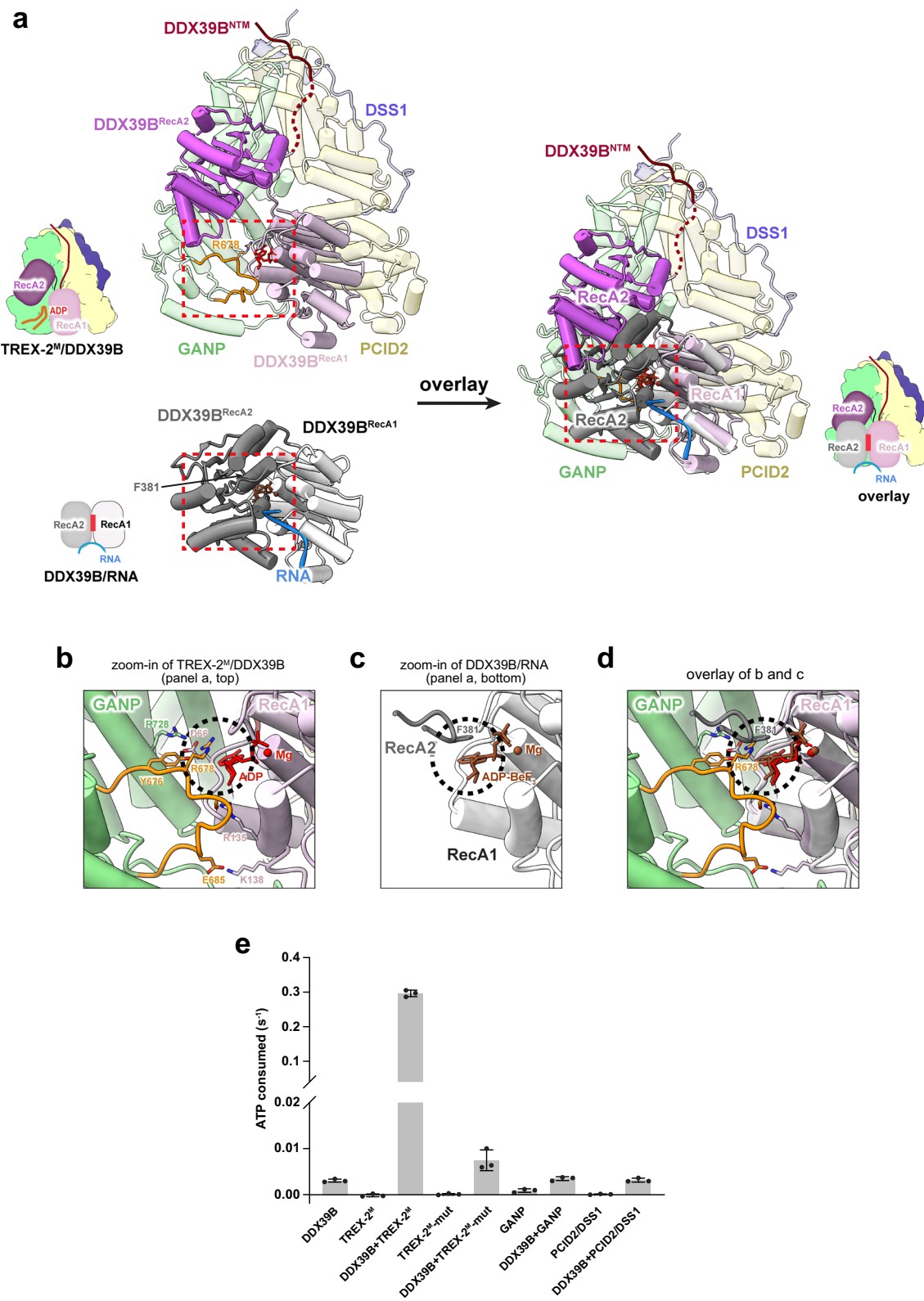

**Fig. 4 | Molecular basis for TREX-2^M-mediated DDX39B regulation. a** Overlay of the overall TREX-2^M/DDX39B structure with the DDX39B/RNA structure[26] (PDB 8ENK) through the RecA1 domain. A model of TREX-2^M/DDX39B was generated with the TREX-2^M/DDX39B^{NTM+RecA1} complex (PDB 9DLP) and docked DDX39B–RecA2 domain based on the cryo-EM map of TREX-2^M/DDX39B (EMD-46981). The trigger loop (colored in orange) from the GANP subunit of TREX-2^M inserts between the two RecA domains. **b–d** The trigger loop directly contacts ADP and breaks key interactions between DDX39B–RecA2 and ADP. Zoom-in views of panel a: TREX-2^M/DDX39B (**b**), DDX39B/RNA (**c**), and their overlay (**d**). **e** The trigger loop and the integrity of the TREX-2^M complex are required for the stimulative effect on DDX39B. Each column represents the mean of 3 independent replicates. Error bars represent standard deviation. Source Data are provided as a Source Data file.

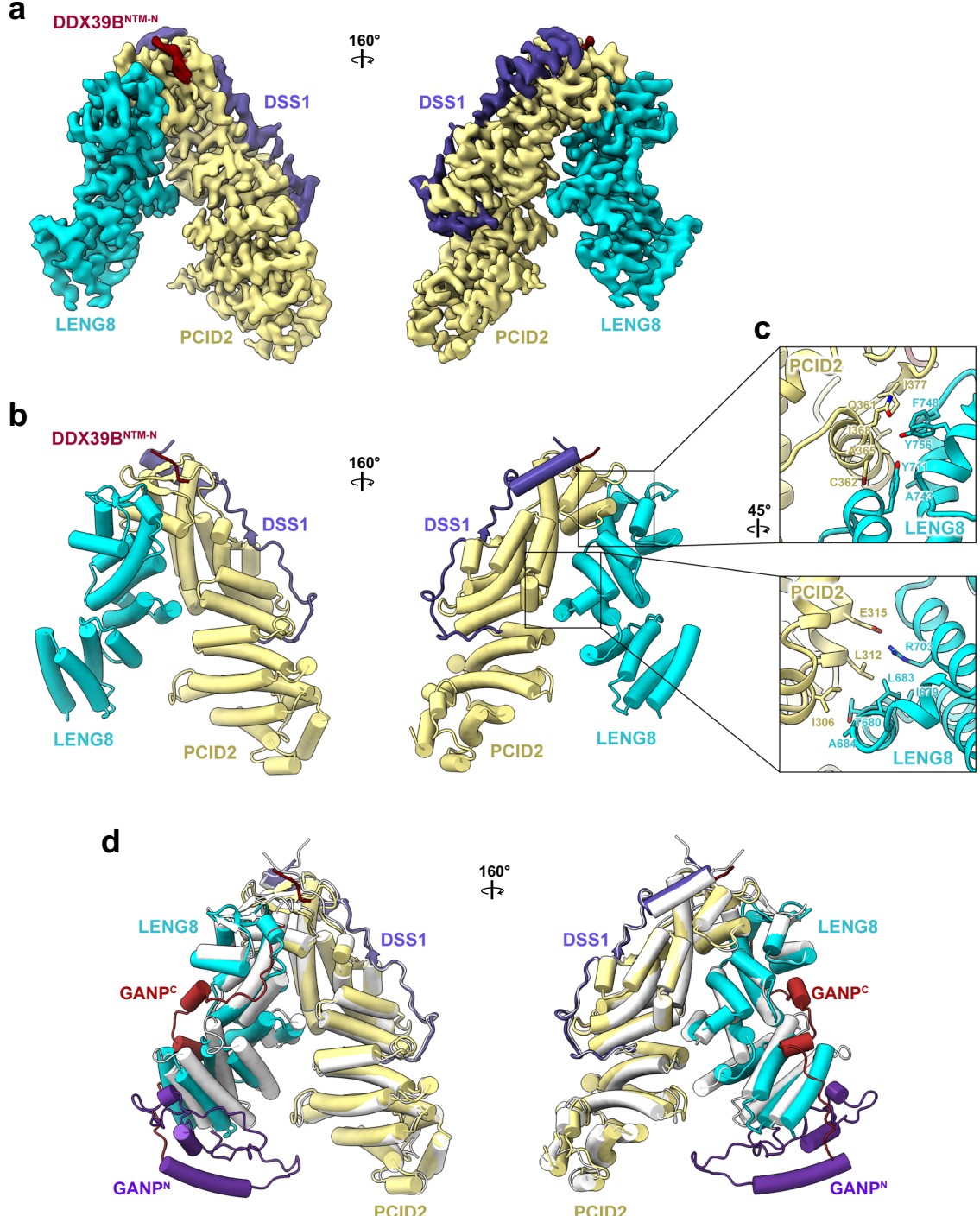

**Fig. 5 | Cryo-EM structure of the human TREX-2.1/DDX39B^NTM-N complex. a** Cryo-EM map of the TREX-2.1/DDX39B^NTM-N complex determined at 3.08 Å resolution (EMD-46983). The LENG8, PCID2, and DSS1 subunits of TREX-2.1 are colored in cyan, yellow, and blue, respectively. The NTM-N of DDX39B is colored in red. **b** Structure of the TREX-2.1/DDX39B^NTM-N complex (PDB 9DLR). **c** Key interfaces between LENG8 and PCID2 in TREX-2.1 assembly. **d** Comparison of TREX-2.1 and TREX-2^M. The TREX-2.1 complex is colored as in (**b**). The TREX-2^M complex is colored white except for the unique N-terminal region (deep purple) and the C-terminal region (red) within GANP.

TREX-2.1, TREX-2.1-mut showed about 4-fold reduction of stimulation on the ATPase activity of DDX39B (Fig. 7c). We showed in Fig. 2c that TREX-2.1/DDX39B can release the ATPase from a pre-assembled DDX39B/RNA complex. TREX-2.1-mut was deficient in this remodeling activity compared to the wild-type TREX-2.1 (Fig. 7d). Thus, our studies reveal the critical role of the conserved LENG8 trigger loop in DDX39B regulation.

## LENG8 (TREX-2.1) influences the nucleocytoplasmic ratio of a subset of mRNAs

To determine the effect of LENG8 on the intracellular distribution of mRNAs, we carried out RNAseq studies on LENG8-depleted cells by siRNAs. RNAs were purified from whole cell lysates, nuclear and cytoplasmic fractions, which were then subjected to RNAseq analysis (Fig. 8a–j, Supplementary Data 1). The observed higher levels of

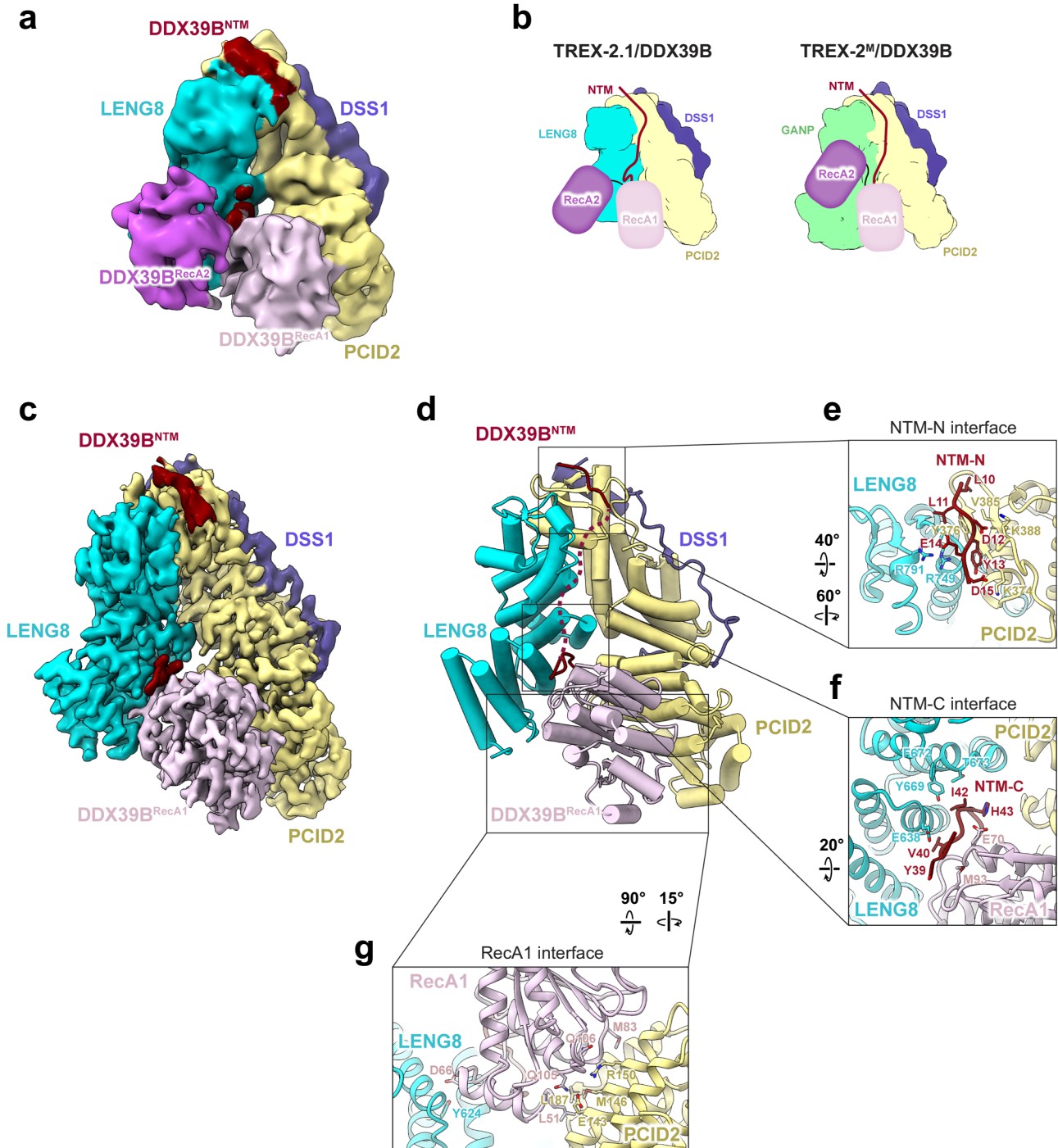

**Fig. 6 | Cryo-EM structure of the human TREX-2.1/DDX39B complex. a** Cryo-EM map of the TREX-2.1/DDX39B complex determined at 3.28 Å resolution (EMD-47126), low-pass filtered at 6 Å resolution. The LENG8, PCID2, and DSS1 subunits of TREX-2.1 are colored in cyan, yellow, and blue, respectively. The NTM, RecA1, and RecA2 domains of DDX39B are colored in red, pink, and purple, respectively. **b** Schematics for TREX-2.1 and TREX-2^M interactions with DDX39B. The LENG8 subunit of TREX-2.1 is shorter than the GANP subunit of TREX-2^M near the DDX39B–RecA1 interface. The RecA2 domain of DDX39B binds to different sites on TREX-2.1 or TREX-2^M. **c** Cryo-EM map of the TREX-2.1/DDX39B^NTM+RecA1 complex determined at 2.97 Å resolution (EMD-46985). **d** Structure of the TREX-2.1/DDX39B^NTM+RecA1 complex (PDB 9DLV). **e–g** TREX-2.1 interfaces with NTM-N (**e**), NTM-C (**f**), and RecA1 (**g**) of DDX39B.

MALAT1 RNA in the nuclear fractions, along with higher levels of GAPDH or actin mRNAs in the cytoplasm than in the nucleus, corroborated the fractionation procedure (Supplementary Data 1, Table 1). Additionally, knockdown of LENG8 mRNA was further corroborated by qPCR (Supplementary Fig. 10a) and at the protein level by western blot (Supplementary Fig. 10b). Upon LENG8 depletion, most cellular mRNAs showed no substantial changes in their total

expression levels. Of 18,995 mRNAs detected by RNAseq analysis, only 1160 mRNAs were up-regulated (>2-fold) and 619 mRNAs were down-regulated (<0.5-fold) in LENG8 knockdown samples compared to control samples, leaving 17,216 mRNAs (~90%) not affected in their total levels (Supplementary Data 1, Tables 2 and 3), indicating that LENG8 depletion did not alter bulk gene expression.

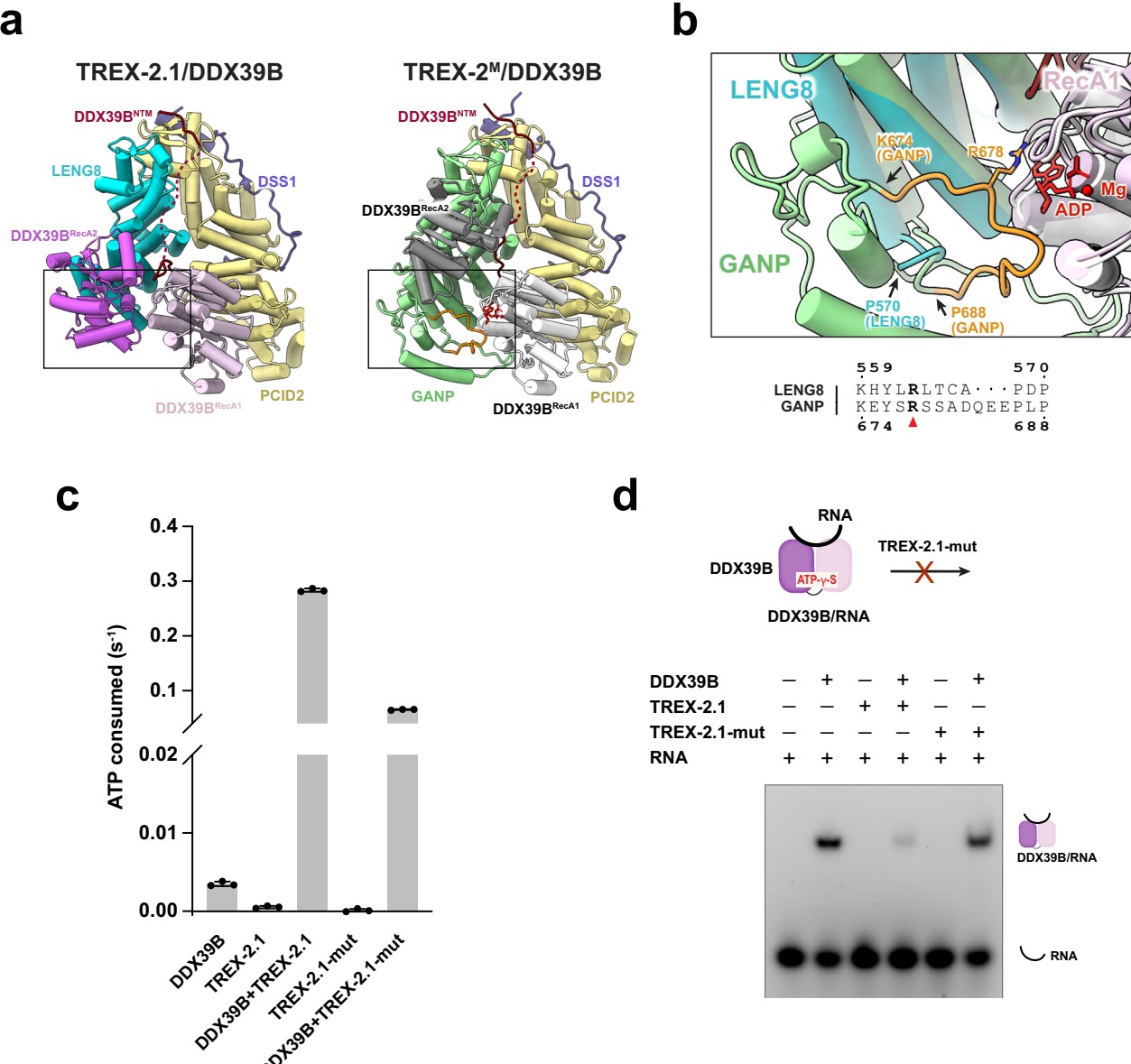

**Fig. 7 | TREX-2.1 regulates DDX39B through the conserved trigger loop.** **a** Structural comparison of TREX-2.1 and TREX-2$^M$ at the trigger loop interface with DDX39B. **b** Zoom-in view of the rectangular region in (**a**) with emphasis on the interaction between the trigger loop and the RecA1 domain of DDX39B. The models of TREX-2.1/DDX39B and TREX-2$^M$/DDX39B were aligned on the RecA1 domain of DDX39B. **c** Trigger loop mutant of TREX-2.1 showed reduced stimulation of the ATPase activity of DDX39B. Each column represents the mean of 3 independent replicates. Error bars represent standard deviation. **d** Trigger loop mutant of TREX-2.1 is deficient in releasing DDX39B from RNA. This experiment has been repeated three times independently with similar results. Source Data are provided as a Source Data file.

To identify mRNAs whose nucleocytoplasmic distribution is modulated by LENG8, we selected mRNAs whose total expression levels were not substantially altered but their nuclear to cytoplasmic (N/C) ratio was 2-fold or more in LENG8 knockdown samples versus controls (Supplementary Data 1, Tables 4 and 5). We identified 391 protein-coding mRNAs retained in the nucleus upon LENG8 depletion (Fig. 8a–j, Supplementary Data 1, Tables 4 and 5). Analysis of shared RNA features of this subset of LENG8-dependent mRNAs (Supplementary Data 1, Table 6) was performed by comparing them to control genome-wide protein-coding mRNAs. In LENG8-depleted cells, mRNAs with an overall high GC content were enriched (Fig. 8a), specifically at their 3'-UTRs and coding sequences (Fig. 8b, c), and less prominent at the 5'-UTR (Fig. 8d). Additionally, LENG8-dependent mRNAs have a slightly higher number of exons (Fig. 8e)

and longer transcript length than control (Fig. 8f). These transcripts also have longer 5'-UTR length (Fig. 8g), slightly shorter mean intron length (Fig. 8h), and the mean exon length as well as the 3'-UTR length are both comparable to control (Fig. 8i, j). Together, our results indicate that LENG8 (TREX-2.1) modulates the nucleocytoplasmic distribution of a subset of mRNAs with high GC content, high exon number, and long transcript length, as compared to the control mRNA population.

**Influenza A virus NP protein competes with TREX-2 or TREX-2.1 for DDX39B binding**

As an important factor in mRNP processing, it is not surprising that DDX39B is targeted and/or hijacked by viruses for their survival advantage[118–120]. We have previously shown that the IAV NP protein

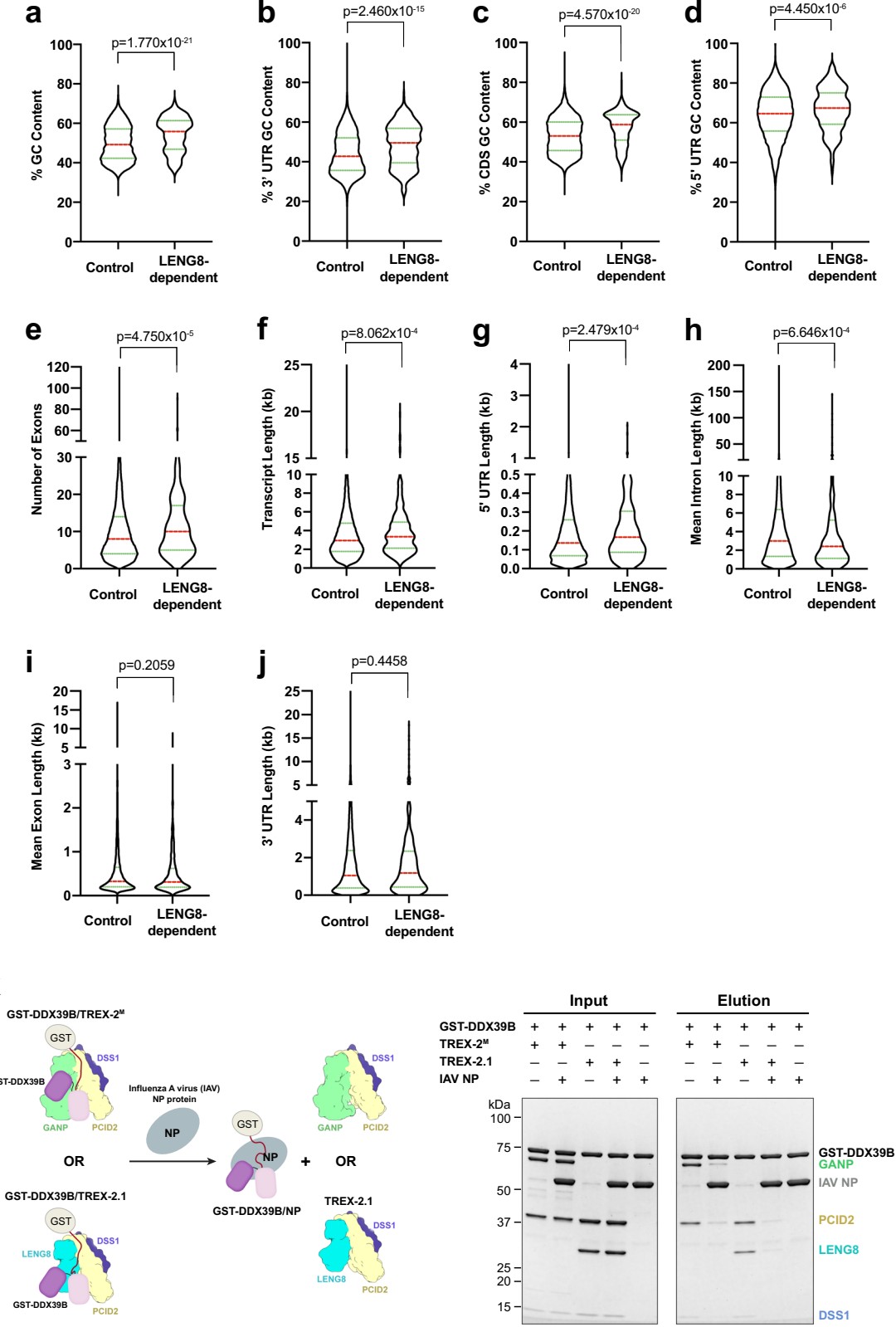

interacts with DDX39B through its NTM and RecA domains[119]. In this study, we reveal that TREX-2 and TREX-2.1 also recognize the NTM and RecA domains of DDX39B. We tested the effect of IAV NP on the TREX-2/TREX-2.1 interactions with DDX39B. Indeed, IAV NP can disrupt the interaction between GST-DDX39B and both the TREX-2/TREX-2.1 complexes in pull-down assays (Fig. 8k). Our studies suggest that IAV NP may target DDX39B and its associated transcription and/or export processes.

## Discussion

We showed that both TREX-2 and TREX-2.1 directly interact with and facilitate the mRNP remodeling activity of DDX39B. The key feature, a conserved trigger loop, is present in both GANP and LENG8 and is crucial for regulating the functional cycle of DDX39B. The snapshots captured from our structures of the TREX-2^M/DDX39B and TREX-2.1/ DDX39B complexes add to our understanding of the full cycle of

**Fig. 8 | Functional investigations of TREX-2 and TREX-2.1 in mRNP metabolism. a–j** RNA features associated with cellular mRNAs whose nucleocytoplasmic distribution is influenced by LENG8. A549 cells were transfected with siRNAs that target LENG8 or with non-targeting control siRNAs. After 72 h, RNA was isolated from whole cell lysates and from nuclear or cytoplasmic fractions and subjected to RNAseq analysis. Two biological replicates were analyzed to identify mRNAs whose nuclear to cytoplasmic ratios are above 2.0 with no change in total expression levels. RNA features associated with cellular mRNAs whose nucleocytoplasmic ratio is modulated by LENG8 were compared to RNA features of the control genome-wide mRNA population. Violin plots show the distribution of the depicted RNA features between these two groups. The red line indicates the median, and the green line indicates quartiles. A two-tailed Mann–Whitney U rank test was used to calculate statistical significance. *p*-values are shown on the top of and the number of transcripts for each group are the following: **a** (control *n* = 19,340; LENG8-dependent *n* = 391), **b** (control *n* = 19,340; LENG8-dependent *n* = 391), **c** (control *n* = 19,340; LENG8-dependent *n* = 391), **d** (control *n* = 19,128; LENG8-dependent *n* = 391), **e** (control *n* = 19,340; LENG8-dependent *n* = 391), **f** (control *n* = 19,340; LENG8-dependent *n* = 391), **g** (control *n* = 19,128; LENG8-dependent *n* = 391), **h** (control *n* = 18,575; LENG8-dependent *n* = 383), **i** (control *n* = 19,340; LENG8-dependent *n* = 391), **j** (control *n* = 19,340; LENG8-dependent *n* = 391). **k** IAV NP disrupts TREX-2/TREX-2.1 interactions with DDX39B. In vitro GST pull-down was performed with purified recombinant proteins. The schematic for the experiment is shown on the left. GST-DDX39B was first preincubated with TREX-2^M or TREX-2.1, and then IAV NP was added to the pull-down reactions. IAV NP was observed to associate with GST-DDX39B and displace TREX-2^M or TREX-2.1, as shown in the right panel. This experiment has been repeated three times independently with similar results. Source Data are provided as a Source Data file.

DDX39B regulation (Fig. 9). During mRNP assembly, THO primes DDX39B/Sub2 in a half-open conformation to facilitate its loading onto mRNPs[24,25] (Fig. 9, step 1). The RNA-bound DDX39B/Sub2 now assumes a closed conformation in association with some mRNP assembly factors, such as SARNP/Tho1 and ALYREF/Yra1[24,26] (Fig. 9, step 2). The TREX-2^M/DDX39B structure traps a state just after ATP hydrolysis while ADP is still present (Fig. 9, step 3). This state reveals that the trigger loop of GANP, especially residue R678, directly engages with ADP and inserts between the two RecA domains of DDX39B, triggering the release of RNA. The TREX-2.1/DDX39B structure captures an open conformation of DDX39B after both ADP and RNA have been released (Fig. 9, step 4). Free DDX39B shows an open conformation (Fig. 9, step 5). During the revision of this manuscript, the structures of yeast TREX-2 in association with Sub2 were published[121]. Compared to the structures of TREX-2/DDX39B and TREX-2.1/DDX39B, the trigger loop within the yeast TREX-2/Sub2 complex adopts a conformation in between, in which the ADP has been released and the trigger loop of TREX-2 is still engaged with Sub2 (Supplementary Fig. 11). A preprint reported an EM reconstruction of TREX-2/DDX39B from a PCID2-DDX39B fusion protein in the presence of AMPPNP and RNA[122]. Consistent with our RNP remodeling assay (Fig. 2d), showing that TREX-2.1 disassembles the RNP composed of AMPPNP, RNA, and DDX39B, RNA is not present in the resulting structure and likely dissociates from DDX39B after TREX-2-mediated remodeling activity. Together, the series of structural snapshots provides mechanistic insights into the ATPase cycle of the crucial DEAD-box ATPase DDX39B.

In human cells, nuclear speckles serve as sites for splicing of a subset of mRNAs, mRNP packaging, and quality control, and they are enriched with RNA processing factors, including DDX39B, THO, ALYREF, and SARNP[123–126]. Structural and biochemical studies suggest that DDX39B, SARNP, and ALYREF facilitate high-order mRNP assembly[24–26]. Release of cellular mRNPs from nuclear speckles requires the activity of DDX39B[67–69]. Our immunofluorescence studies show that LENG8 is localized to the nucleus (Fig. 1d).

Thus, TREX-2.1 likely regulates the function of DDX39B inside the nucleus, although the cellular context of their interaction remains to be studied. In line with its nuclear localization, LENG8 associates with factors involved in nuclear mRNA biogenesis at various steps. One group of LENG8-associated proteins plays roles primarily in mRNP assembly, such as TREX, TREX-2, ZC3H11A, and U2AF1[98,99,101,102]. Another group of LENG8-associated factors, such as ZC3H18, ZFC3H1, and components of the exosome complex, are involved in mRNP quality control[57,97,101,102].

It should be noted that most mRNP processing factors discussed above are multi-functional. For instance, Gbp2 and ZC3H14/Nab2 are involved in both mRNP packaging and quality control[25,43,53–55], and often their functions are regulated by post-transcriptional modifications, such as phosphorylation on yeast Gbp2 and human ZC3H14[127,128]. These connections highlight that mRNP biogenesis and degradation processes need to be kept in

balance[5]. It would be interesting to investigate how TREX-2.1 and the population of nuclear-localized TREX-2 facilitate co-transcriptional mRNP assembly inside the nucleus. The majority of TREX-2 localizes at the NPC, coupling the final steps of mRNP assembly for the export of mature mRNPs. TREX-2 directly interacts with both DDX39B and NXF1-NXT1, which may facilitate DDX39B removal as well as NXF1-NXT1 loading onto mRNPs at the NPC. In vivo, mRNPs are highly dynamic, and their assembly involves a large number of biogenesis factors[4,40,41,129]. The coordinated functions of these factors warrant further study.

Our RNAseq analysis reveals that mRNAs whose nuclear to cytoplasmic ratio is modulated by LENG8 have higher GC content at the 3′-UTR and coding regions. It has been shown that the RBM33 protein interacts with GC stretches and recruits the mRNA nuclear export receptor NXF1[130]. Our data suggest that the high GC content in LENG8-dependent mRNAs might facilitate their nuclear export, although it remains possible that potential defects in other RNA processes may contribute to the observed nucleocytoplasmic distribution. In our previous study, using the auxin-inducible degron system to deplete GANP, we found that GANP-dependent mRNAs have an overall lower GC content than the GANP-independent mRNAs[131], which suggests that the TREX-2.1 complex may mediate nuclear export of a subset of mRNAs with high GC content that are distinct from GANP-dependent mRNAs. Additionally, LENG8-dependent mRNAs have a slightly higher number of exons, longer transcripts, and 5′-UTR length, and shorter mean intron length. These features suggest that LENG8 is involved in mediating nuclear export of mRNAs that undergo more splicing events, a process known to facilitate recruitment of mRNA export factors[59].

mRNP export factors including DDX39B, TREX-2, and NXF1-NXT1 are extensively exploited by viruses. IAV mRNA hijacks the cellular mRNP assembly machinery. IAV M mRNA transits through nuclear speckles, and DDX39B supports M mRNA processing at the nuclear speckles[67]. In addition, the interplay between cellular TREX-2, NXF1-NXT1, and NS1-BP proteins and IAV NS1 protein is required for IAV mRNA export[67,131–133]. NXF1-NXT1, as the principal mRNP export receptor, is targeted by IAV NS1 protein and SARS-CoV-2 Nsp1 protein to inhibit host mRNA nuclear export and suppress host immune response by blocking the expression of antiviral factors[134–137]. Interestingly, NXF1-NXT1 is also hijacked by retroviral CTE-RNA to bypass cellular quality control mechanisms for exporting unspliced transcripts to the cytoplasm[138–140]. Viral factors that target host mRNA nuclear export are not only potential targets for antiviral development but can also be powerful tools for understanding cellular mechanisms. In this study, we discover that the IAV NP protein directly targets the TREX-2/TREX-2.1 and DDX39B interactions (Fig. 8k). This finding could be used to probe the cellular pathways involving DDX39B, THO, TREX-2, TREX-2.1, and NXF1-NXT1 in future studies.

Together, we suggest that DDX39B and associated factors regulate nuclear mRNP assembly at multiple levels: splicing steps with

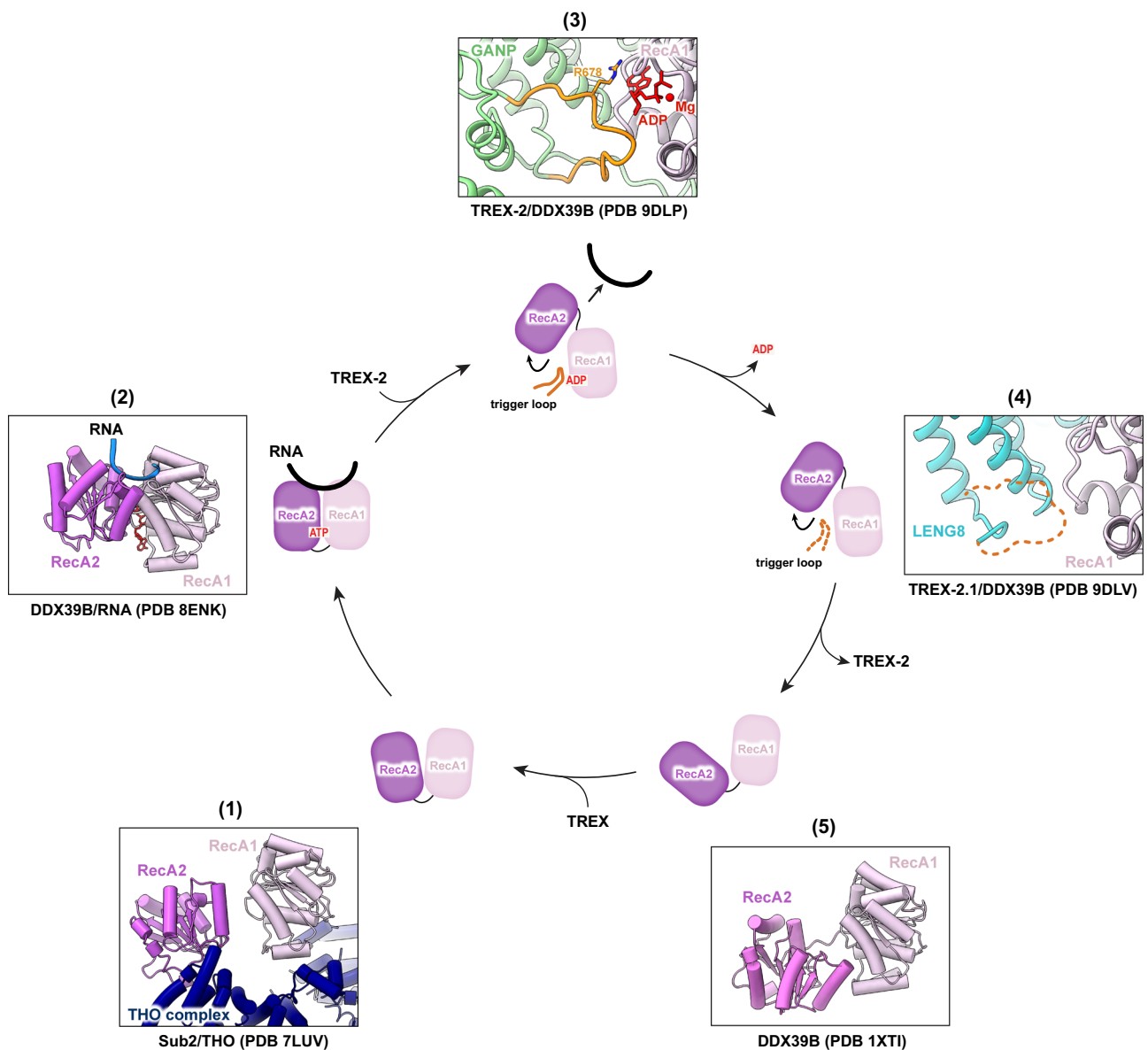

**Fig. 9 | Functional cycle of DDX39B and its regulation by TREX-2 and TREX-2.1.** Schematic of the DDX39B ATPase functional cycle with representative structures of each state. (1) The THO complex primes DDX39B in a semi-open conformation and recruits the ATPase to nascent transcripts[24,25]. (2) DDX39B assumes a closed conformation when bound to RNA[24,26]. (3) Structure of TREX-2/DDX39B captures a key intermediate, in which RNA has been released and ADP is still engaged by the critical trigger loop from TREX-2. (4) Structure of TREX-2.1/DDX39B provides a snapshot after both RNA and nucleotide have been released from DDX39B. (5) DDX39B adopts an open conformation in the absence of nucleotide and RNA[150,151].

U2AF1/U2AF2; mRNP packaging steps with THO/ALYREF/SARNP; mRNP remodeling inside the nucleus with TREX-2.1 and nuclear-localized TREX-2; and final mRNP export with NPC-associated TREX-2. The spatial and temporal regulation of these multi-functional factors is important to better understand their roles. In this study, the separation of function mutants identified on the trigger loops of TREX-2 and TREX-2.1, critical for DDX39B regulation, will be valuable to dissect the highly integrated and coordinated nuclear mRNP biogenesis and export pathways.

## Methods
### Plasmids
GST-tagged constructs including human LENG8 (residues 490–800) wild type or mutant ($^{563}$RLTCAP$^{568}$ replaced with GGGG), human LENG8 (residues 549–800) wild type or mutant ($^{563}$RLTCAP$^{568}$ replaced with GGGG), human GANP (residues 358–1000) wild type or mutant ($^{678}$RSSADQEEP$^{686}$ replaced with GGGG), human GANP (residues 581–1000), and human DDX39B (residues 1–428) were cloned into a pGEX-4T-1 vector modified to contain a TEV cleavable GST tag. His-tagged (TEV cleavable) human PCID2 (residues 1–399) and DSS1 (residues 1–70) were cloned into the MCS1 and MCS2 of pRSFDuet-1, respectively. His-tagged human DDX39B (residues 1–428) was cloned in pProEx-HTb. Human THOC1 (residues 1–657), His-THOC5 (residues 12–230), and strep-THOC7 (residues 1–204) were cloned into a pACEBac1 vector. Human His-THOC2 (residues 1–1180) and Flag-THOC3 (residues 1–351) were cloned into a pFastBac Dual vector. For generating pBpa incorporated DDX39B used for cryo-EM studies, residue 108 of the GST-DDX39B (residues 1–428) plasmid was replaced by the amber stop codon TAG. For immunoprecipitation experiments, full-length LENG8 was cloned into a pCMVTnT vector modified to contain an N-terminal 3xFlag tag, and full-length PCID2 was cloned into a pCMVTnT vector modified to contain an N-terminal 3xHA tag. Protein mutations were introduced using the NEBuilder HiFi DNA assembly cloning kit (NEB).

## Cell culture

Human lung adenocarcinoma epithelial cells (A549, CCL-185, ATCC) or 293T (CRL-3216, ATCC) cells were maintained in high-glucose DMEM media (Gibco) supplemented with 10% FBS (Sigma) and 100 units/ml Pen/Strep antibiotics at 37 °C with 5% $CO_2$.

## Immunofluorescence microscopy

A549 cells were grown on UV-sterilized, gelatin-coated 12 mm coverslips in a 24-well plate at a density of 50,000 cells per well, in 80% (v/v) complete media (91% (v/v) DMEM + 9% (v/v) FBS) with 20% (v/v) OptiMEM I (#31985070, Gibco) for 72 h at 37 °C.

After removing cell culture media and rinsing in PBS, cells were fixed for 20 min in 4% paraformaldehyde (in PBS) and then rinsed briefly in PBS. Cells were permeabilized for 10 min with 0.3% (v/v) Triton X-100 in PBS and then rinsed briefly in PBS. The cells were blocked for 60 min in 5% (w/v) BSA with 0.1% (v/v) Triton X-100 in PBS.

Primary antibody incubation was performed for 60 min at 37 °C. Each coverslip was placed cells-down on a 36 μL drop of primary antibody solution (3% (w/v) BSA fraction V with 0.1% (v/v) Triton X-100 in PBS), containing rabbit anti-LENG8 (#HPA061571, Sigma-Aldrich) at a dilution of 1:50 (1.0 ng/μL) and MAB414 monoclonal mouse anti-NPC (#N8786-100UL, Sigma-Aldrich) at a dilution of 1:1000 (3 ng/μL). After incubation, the coverslips were returned to the 24-well plate and washed in 0.1% (v/v) Triton X-100 in PBS twice for 5 min each.

Secondary antibody incubation was performed for 60 min at 37 °C. Each coverslip was placed cells-down on a 36 μL drop of secondary antibody solution (3% (w/v) BSA fraction V with 0.1% (v/v) Triton X-100 in PBS) containing goat anti-rabbit with Alexa Fluor 488 (#A32731, Invitrogen) and goat anti-mouse with Alexa Fluor 546 (#A11030, Invitrogen), each at a dilution of 1:300 (6.7 ng/μL). After incubation, the coverslips were returned to the plate, washed in 0.1% (v/v) Triton X-100 in PBS twice for 5 min each, and rinsed once in autoclaved milli-Q water.

DNA staining was performed using a 1:10,000 solution (1.0 ng/μL) of Hoechst 33258 (bis-benzimide, #H-3569, Molecular Probes) in autoclaved milli-Q water for 5 min. After DNA staining, each coverslip was briefly rinsed once in autoclaved milli-Q water, then mounted on a slide with 9 μL of Prolong Gold antifade (#P36930, Invitrogen) and cured overnight. Cells were imaged using a Zeiss Axiovert 200 M inverted widefield fluorescence microscope. Images were deconvolved by a blind 3D algorithm in the AutoQuant X3 software. Representative images were produced in Fiji 2.9.0, using a single in-focus z plane. Scale bars were added using Fiji.

## Immunoprecipitation

HA-PCID2 plasmid was co-transfected with Flag vector control or Flag-LENG8 in 293T cells by lipofectamine 2000 (Invitrogen) in a 6-well plate for 48 h according to the manufacturer's protocol. $6 \times 10^6$ cells were suspended in 500 μL of ice-cold lysis buffer (50 mM Tris- HCl pH 7.5, 150 mM NaCl, 1% IGEPAL CA-630, 0.1 mM $Na_3VO_4$, 1 mM NaF, 1 mM DTT, 1 mM EDTA, 1 mM PMSF, 2× cOmplete protease inhibitor mixture, and 10% glycerol). The cells in lysis buffer were incubated on ice for 10 min, and during this step, cells were suspended by inverting every 5 min. Next, cell lysates were subjected to sonication on ice and centrifuged for 15 min at 4 °C at $15,000 \times g$. The supernatant was applied to Anti-FLAG M2 magnetic beads (Sigma-Aldrich) for binding at 4 °C for 2 h. Beads were washed five times with lysis buffer at 4 °C. Then proteins were eluted using 10 μg/mL 3×Flag peptide dissolved in PBS. The eluted proteins were mixed with sample buffer, boiled, and subjected to SDS-PAGE. Mouse anti-Flag (1:1000 dilution, F1804, Sigma), rabbit anti-HA (1:500 dilution, 3724, Cell Signaling Technology), and mouse anti-UAP56 (1:500 dilution, MA5-38508, Invitrogen) were used for western blot. Anti-mouse IgG HRP (1:7000 dilution, NA931, Cytiva) and anti-rabbit IgG HRP (1:7000 dilution, 31460, Invitrogen) were used as secondary antibodies.

## RNA interference and transfection

A pool of 4 siRNAs that target LENG8 (PreDesigned siRNA, NM_052925, Sigma-Aldrich) was used for RNA knockdown of LENG8. Non-targeting siRNAs (Mission siRNA Universal Negative Control, SIC001-10NMOL, Sigma-Aldrich) were used as controls. siRNAs were used at a final concentration of 100 nM. A549 cells were reverse-transfected with siRNAs using RNAiMAX (Invitrogen) according to the manufacturer's instructions. After 72 h, cells were lysed for biochemical approaches.

## Western blot

For Supplementary Fig. 10, cells were lysed in sample buffer (63 mM Tris pH 7.0, 10% glycerol, and 2% SDS in nuclease-free water). Cell lysates were heated at 95 °C for 10 min and sonicated briefly. Cell debris was removed by centrifugation at $13,800 \times g$ for 5 min at 4 °C. The supernatant was subjected to 8% SDS-PAGE in a running buffer (Invitrogen, Cat. B0002-02), followed by western blot. Proteins were transferred to a PVDF membrane with 0.45 μm pore size (Millipore, Cat. IPVH00010) in a transfer buffer (Bio-Rad, Cat. 161-0771). The membrane was blocked with 5% nonfat milk in TBS-T buffer (Thermo Fisher, Cat. 28360) supplemented with 1% bovine serum albumin (HyClone, Cat. SH30574.02) at room temperature for 1 h. Primary (LENG8 polyclonal antibody, 1:500 dilution, PA5-116606, Invitrogen) and secondary (anti-rabbit IgG HRP, 1:7000 dilution, 31460, Invitrogen) antibodies were diluted with TBS-T supplemented with 1% BSA. Incubation of antibodies was performed for 1 h at room temperature. The membrane was washed 5 times for 5 min each time with TBS-T. The LI-COR Odyssey imaging system was used to detect the chemiluminescence signal.

## Cell fractionation, RNA purification, and quantitative RT-PCR

Cells were harvested by trypsinization and centrifugation. The cell pellets were washed once with cold PBS and pelleted at $900 \times g$ for 5 min at 4 °C. The cell pellets were resuspended with 1 mL RSB buffer (10 mM Tris pH 7.4, 10 mM NaCl, 3 mM $MgCl_2$), incubated for 3 min on ice, and then centrifuged at $900 \times g$ for 5 min at 4 °C. The volume of the pelleted cells was estimated, and four times the volume of lysis buffer RSB40 (10% glycerol, 0.5% IGEPAL CA-630, 0.5 mM DTT, and 100 U/mL RNasin rnase inhibitor added in RSB buffer) was used to resuspend the pellet by slow pipetting. Nuclei were pelleted by centrifugation at $4700 \times g$ for 3 min, and the supernatant was recovered and saved as the cytoplasmic fraction. Nuclear pellets were resuspended in RSBG40, and one-tenth volume of detergent (3.3% (wt/wt) sodium deoxycholate and 6.6% (vol/vol) Tween 40) was added with slow vortexing, followed by incubation on ice for 5 min. Nuclei were again pelleted, and the supernatant was pooled with the previous cytoplasmic fraction. Nuclear pellets were washed once in RSBG40, collected at $9600 \times g$ for 5 min, and the resulting pellet was used for nuclear RNA extraction. RNA was isolated with the RNeasy Plus Mini Kit (Qiagen). cDNA was made with the iScript cDNA synthesis kit (Bio-Rad) from 0.8 μg RNA and diluted 1:5 with nuclease-free water, followed by quantitative PCR using the LightCycler 480 SYBR green system (Roche) according to the manufacturer's instructions. 18S rRNA was used as an internal control. The primers used for RT-PCR are listed below.

LENG8qPCRFor: 5'-CACACACCTACACCGAACCTG-3'
LENG8qPCRRev: 5'-AAAGGGTCGCTTCTGGATGTT-3'
18SFor: 5'-ACCGCAGCTAGGAATAATGG-3'
18SRev: 5'-GCCTCAGTTCCGAAAACCA-3'

## mRNAseq

RNA Samples from total cell lysates, nuclear (N) or cytoplasmic (C) fractions, were run on the Agilent Tapestation 4200 to determine the level of degradation, thus ensuring that only high-quality RNA is used (RIN Score 8 or higher). The Qubit fluorimeter was used to determine the concentration prior to starting library preparation. One microgram

of total DNase-treated RNA was then prepared with the TruSeq Stranded mRNA Library Prep Kit from Illumina. Poly(A) RNA was purified and fragmented before strand-specific cDNA synthesis. cDNAs were then A-tailed, and indexed adapters were ligated. After adapter ligation, samples were PCR amplified and purified with AmpureXP beads, then validated again on the Agilent Tapestation 4200. Before being normalized and pooled, samples were quantified by Qubit and then run on the Illumina NextSeq 2000 using the NextSeq™ 1000/2000 P2 XLEAP-SBS™ Reagent Kit (300 Cycles).

### RNAseq data analysis

Raw sequence data was trimmed using Trimmomatic[141]. QC filtered trimmed sequences were aligned to hg19 using STAR[142]. All subsequent analysis was performed using R version 4.0.2 and Bioconductor 3.11 in RStudio[143]. Raw counts were obtained from BAM files using Feature-Counts from the Rsubread package[144], and TPM was calculated from raw counts. TPM values were used for calculating the N/C ratio of mRNAs for control and LENG8 knockdown samples. Relative change in the N/C ratio was calculated by comparing the LENG8 knockdown to the control sample. Transcripts whose total levels were not substantially altered (between 0.5-fold and less than 2-fold) and with a relative change in N/C ratio ≥2 in two independent experiments were considered for further analysis. RNA Features (transcript length, 5′UTR length, 3′UTR length, exon count, mean exon length, mean intron length, %GC content of full-length mRNA, 5′UTR, CDS, and 3′UTR) of transcripts retained in the nucleus were extracted based on Ensembl GTF annotations (GRCh38, release version 104) for each Ensembl Canonical transcript. RNA features of transcripts belonging to protein-coding biotypes were selected. Violin plots were generated to compare the distribution of these RNA features between transcripts retained in the nucleus upon LENG8 depletion versus RNA features of control genome-wide protein-coding mRNAs using GraphPad Prism 9. Statistical significance was assessed using a two-tailed Mann–Whitney U rank test using GraphPad Prism 9.

### Protein expression and purification

DDX39B, TREX-2, and TREX-2.1 proteins were expressed in *E. coli* Rosetta cells (Sigma-Aldrich), except for the *p*Bpa-incorporated DDX39B protein. Protein expression in Rosetta cells was induced by 0.5 mM IPTG at 20 °C overnight. Cells were lysed in a buffer containing 50 mM Tris, pH 8.0, 300 mM NaCl, 0.5 mM TCEP, 0.1 mM AEBSF, and 2 mg/L aprotinin.

TREX-2.1 (LENG8/PCID2/DSS1) and TREX-2^M (GANP/PCID2/DSS1) complexes were obtained by co-expressing the plasmids encoding GST-LENG8 (residues 490–800, or residues 549–800) or GST-GANP (residues 358–1000) and the His-PCID2/DSS1 complex. Samples were subjected to tandem affinity purification, first with Ni-Sepharose 6 Fast Flow resin (Cytiva) and subsequently with Glutathione Sepharose 4B resin (Cytiva) to obtain a stoichiometric complex. Next, expression tags were cleaved with GST-TEV at 4 °C overnight. Digested proteins were purified on a Source 15Q column (Cytiva) and were further purified on a Superdex 200 column (Cytiva) equilibrated with 10 mM Tris pH 8.0, 150 mM NaCl, and 0.5 mM TCEP.

Subunits/subcomplexes of TREX-2 or TREX-2.1 were obtained from the plasmids encoding GST-GANP (residues 581–1000), GST-LENG8 (residues 549–800), or His-PCID2/DSS1. Samples were first purified with Glutathione Sepharose 4B resin (for GST-GANP or GST-LENG8) or Ni-Sepharose 6 Fast Flow resin (for His-PCID2/DSS1), followed by ion exchange chromatography (HiTrap Heparin for GST-LENG8; Source 15Q for GST-GANP or His-PCID2/DSS1). For purification of GST-LENG8, the sample was applied to a Superdex 200 column equilibrated with 10 mM Tris pH 8.0, 300 mM NaCl, and 0.5 mM TCEP. For purification of untagged proteins, GST-TEV (for GST-LENG8 or GST-GANP) or His-TEV (for His-PCID2/DSS1) were added to proteins for tag cleavage at 4 °C overnight. Digested proteins were loaded on

Glutathione Sepharose 4B resin (for LENG8 or GANP) or Ni-Sepharose 6 Fast Flow resin (for PCID2/DDS1) to remove uncleaved proteins, cleaved tag, and TEV. Samples were purified on a Superdex 200 column equilibrated with 10 mM Tris pH 8.0, 300 mM NaCl, and 0.5 mM TCEP.

GST-tagged DDX39B proteins were pulled down using Glutathione Sepharose 4B resin. Proteins were then purified on a Source 15Q column and were further purified on a Superdex 200 column equilibrated with 10 mM Tris pH 8.0, 150 mM NaCl, and 0.5 mM TCEP.

His-tagged DDX39B was pulled down using Ni-Sepharose 6 Fast Flow resin. The His-tag was cleaved with His-TEV at 4 °C overnight. Digested protein was purified on a Source 15Q column and was further purified on a Superdex 200 column (Cytiva) equilibrated with 10 mM Tris pH 8.0, 150 mM NaCl, and 0.5 mM TCEP.

Human THO complex was expressed in High-Five insect cells by coinfection of recombinant baculoviruses generated from the above-mentioned pACEBac1 and pFastBac Dual plasmids. The sample was first affinity-purified with Ni-Sepharose resin, followed by Anti-FLAG M2 affinity resin (Sigma). The protein was further purified using a Q-Sepharose Fast Flow column (Sigma) and a Superose 6 size exclusion column (Cytiva) equilibrated with 10 mM Tris pH 8.0, 300 mM NaCl, 0.5 mM TCEP.

*p*Bpa incorporated DDX39B at residue 108 (DDX39B^{*p*Bpa108}) was expressed using amber codon suppression[109]. The DDX39B^{*p*Bpa108} expression plasmid was co-transformed with an aminoacyl-tRNA synthetase/tRNA pair encoded by a plasmid pEVOL-pBpF (Addgene, 31190) in *E. coli* BL21 (DE3) (NEB). Arabinose was added to the cell culture at a final concentration of 0.02% (w/v) when the optical density (OD) at 600 nm reached 0.3 to induce expression of the aminoacyl-tRNA synthetase/tRNA pair. Cells were spun down when the OD at 600 nm reached 0.6 and resuspended in a volume of media 5-fold less than the initial culture volume in the presence of 1 mM *p*Bpa (AEchem) and 0.02% (w/v) arabinose. Cells were incubated at 16 °C for 1 h, and protein expression was then induced with 0.5 mM IPTG at 16 °C for 16 h. The protein was pulled down using Glutathione Sepharose 4B resin, followed by GST tag cleavage by GST-TEV. Protein was then purified on a Source 15Q column and was further purified on a Superdex 200 column equilibrated with 10 mM Tris pH 8.0, 300 mM NaCl, and 0.5 mM TCEP.

Recombinant influenza virus A/PR/8/34 NP protein was purified as previously described[119].

### Cryo-EM sample preparation and data collection

Stable protein complexes of human TREX-2.1 or TREX-2^M in association with DDX39B for cryo-EM studies were obtained through photo-crosslinking mediated by an unnatural amino acid *p*Bpa incorporated in DDX39B. 5 μM of human TREX-2.1 or TREX-2^M was incubated with 5 μM of DDX39B^{*p*Bpa108} in a buffer containing 10 mM Tris pH 8.0, 60 mM NaCl, 0.5 mM TCEP on ice for 30 min. The protein mixture was irradiated at 365 nm (Spectrolinker) on ice in a 96-well plate (Thermo Fisher Scientific) for 40 min at 2 min intervals, yielding photo-crosslinked TREX-2.1/DDX39B or TREX-2^M/DDX39B complexes. The crosslinked samples were diluted to 3 μM and adjusted salt concentration to 150 mM NaCl. For TREX-2^M/DDX39B, 1 mM ADP and 1 mM MgCl2 were also added to the sample. The samples were deposited on glow-discharged UltrAuFoil R 1.2/1.3 grids (Quantifoil). Grids were blotted for 6 s with a blotting force of 6 at 4 °C and 100% humidity and plunged into liquid ethane using a FEI Vitrobot Mark IV (Thermo Fisher). The data were collected with a Titan Krios G4 (Thermo Fisher) equipped with a K3 direct electron detector (Thermo Fisher). Movies were collected with EPU at a magnification of 105,000×, corresponding to a calibrated pixel size of 0.820 Å/pixel. A total of 16,147 movies were collected with a defocus range from 0.8 μm to 2.0 μm for the TREX-2.1/DDX39B complex. A total of 32,206 movies were collected with a defocus range from 0.8 μm to 2.0 μm for the TREX-2^M/DDX39B

complex. A full description of the cryo-EM data collection parameters can be found in Table 1.

## Cryo-EM image processing and model building of TREX-2$^M$/DDX39B

Cryo-EM data were processed with CryoSPARC (version 4.4)[145]. 32,206 movies were gain-normalized, aligned, and dose-weighted using patch motion correction, followed by patch CTF estimation. A subset of 1000 micrographs was subjected to blob particle picking followed by 2D classification to obtain particles for Topaz (version 0.2.5) training. 22,639,463 particles were picked by Topaz from the entire data set. 2D classification followed by ab initio reconstruction and heterogeneous refinement resulted in 5,437,549 particles that show well-resolved TREX-2$^M$ densities and additional densities corresponding to the RecA domains of DDX39B. Another round of ab initio reconstruction and heterogeneous refinement resulted in a class of 1,044,906 particles that show densities for both RecA1 and RecA2 domains of DDX39B (TREX-2$^M$/DDX39B) and a class of 1,800,272 particles that show the best-resolved RecA1 domain of DDX39B (TREX-2$^M$/DDX39B$^{NTM+RecA1}$). For TREX-2$^M$/DDX39B, further heterogeneous refinement and 3D classification yielded a set of 199,374 particles, resulting in a map refined to 3.25 Å resolution (EMD-46981). For TREX-2$^M$/ DDX39B$^{NTM+RecA1}$, further 2D classification, heterogeneous refinement, and 3D classification yielded a set of 720,007 particles. The final TREX-2$^M$/ DDX39B$^{NTM+RecA1}$ map was refined to 2.79 Å resolution (EMD-46982).

An AlphaFold 2[146] predicted TREX-2$^M$ structure, and the crystal structure of DDX39B (PDB ID 8ENK) was used to generate the initial model for TREX-2$^M$/DDX39B$^{NTM+RecA1}$. The model was manually adjusted in Coot (version 0.9.8)[147] and refined in Phenix (version 1.21.1) using real-space refinement[148]. Refinement statistics of TREX-2$^M$/DDX39B$^{NTM+RecA1}$ (PDB 9DLP) can be found in Table 1.

## Cryo-EM image processing and model building of TREX-2.1/DDX39B

Cryo-EM data were processed with CryoSPARC (version 4.4)[145]. 16,147 movies were gain-normalized, aligned, and dose-weighted using patch motion correction, followed by patch CTF estimation. A subset of 1000 micrographs was subjected to blob particle picking followed by 2D classification to generate 2D class averages for template picking. 19,092,422 particles were picked using Template Picker. Initial processing using ab initio reconstruction and multiple rounds of heterogeneous refinement yielded two maps that both feature the V-shaped architecture of TREX-2.1, and one of the maps additionally shows densities for RecA domains of DDX39B. These two maps were used as reference maps for heterogeneous refinement using all particles, yielding a class of 1,300,367 particles that corresponds to TREX-2.1/DDX39B$^{NTM-N}$ and a class of 2,012,259 particles that corresponds to TREX-2.1/DDX39B. For TREX-2.1/DDX39B$^{NTM-N}$ particles, a further round of heterogeneous refinement, non-uniform refinement, 3D classification, and local refinement yielded a map refined to 3.08 Å resolution with 569,284 particles (EMD-46983). For TREX-2.1/DDX39B particles, further heterogeneous refinement generated a class of 1,160,020 particles that show strong densities for both RecA domains of DDX38B. 3D classification with a focused mask covering the RecA domains generated a set of 289,563 particles, resulting in a TREX-2.1/DDX39B map refined to 3.28 Å resolution (EMD-47126). The TREX-2.1/DDX39B particles also generated a set of 557,019 particles that feature the best densities for the RecA1 domain of DDX39B through 3D classification with a focused mask covering TREX-2.1 and the RecA1 domain. Further non-uniform refinement and local refinement resulted in a TREX-2.1/ DDX39B$^{NTM-RecA1}$ map refined to 2.97 Å resolution (EMD-46985).

An AlphaFold 2[146] predicted TREX-2.1 structure, and the crystal structure of DDX39B (PDB ID 8ENK) was used to generate the initial models for TREX-2.1/DDX39B$^{NTM+RecA1}$ and TREX-2.1/DDX39B$^{NTM-N}$. The

models were manually adjusted in Coot[147] and were refined in Phenix (version 1.21.1) using real-space refinement[148]. Refinement statistics of TREX-2.1/DDX39B$^{NTM+RecA1}$ (PDB 9DLV) and TREX-2.1/DDX39B$^{NTM-N}$ (PDB 9DLR) can be found in Table 1.

## ATPase assay

ATPase activity was analyzed using an NADH enzyme-coupled absorbance assay[149]. Briefly, standard ATPase reactions were prepared with 1.5 μM of indicated proteins (DDX39B; TREX-2$^M$, GANP$_{358-1000}$/PCID2/DSS1, wild type or mutant; TREX-2.1, LENG8$_{490-800}$/PCID2/DSS1, wild type or mutant; GANP$_{581-1000}$; PCID2/DSS1) in 10 mM Tris (pH 8.0), 100 mM (for Fig. 4e) or 70 mM (for Figs. 2d, 7b) NaCl, 2 mM MgCl$_2$, 0.5 mM TCEP, 0.125 mg/mL poly(A) RNA (Cytiva), 1 mM ATP, 5 mM phosphoenolpyruvate, 1.2 mM (reactions with TREX-2$^M$, TREX-2.1, or TREX-2.1-mut) or 0.1 mM (all other reactions) NADH, and 2% (vol/vol) pyruvate kinase/lactate dehydrogenase (Sigma). UV absorbance at 340 nm was monitored by a BioTek Synergy HTX microplate reader at 37 °C. Reaction rates were calculated in GraphPad Prism 10 from the slopes of the linear phase showing the decrease in NADH absorbance as a function of time.

## Electrophoretic mobility shift assay

An EMSA was used to examine the effect of TREX-2.1 on the pre-assembled DDX39B/RNA complex. Poly(U) 10-mer RNA (40 nM) labeled with Alexa Fluor 488 at the 5′ end was mixed with DDX39B (0.6 μM) in a buffer containing 20 mM HEPES (pH 7.0), 100 mM NaCl, 1 mM MgCl$_2$, 6% glycerol, 1 mM ATP-S or 1 mM AMPPNP as indicated, 0.5 mM TCEP, 30 μg/ml BSA, and 0.5 U/μl SUPERase•In RNase Inhibitor (Thermo Fisher Scientific). The mixtures were incubated at room temperature for 10 min. Next, 0.6 μM of the TREX-2.1 complex (LENG8$_{549-800}$/PCID2/DSS1, wild type or mutant) or an individual component (LENG8$_{549-800}$ or PCID2/DSS1) as indicated was added into the above mixture and incubated at room temperature for another 20 min. Samples were separated on a 5% native PAGE gel in 0.5× TB, pH 8.0, running buffer at 4 °C. RNA was visualized with a ChemiDoc MP Imaging System (Bio-Rad).

## GST pull-down

For Fig. 1b and Supplementary Fig. 1a, 1 μM of GST-LENG8$_{549-800}$ was mixed with purified THO complex or PCID2/DSS1 in a binding buffer containing 10 mM Tris pH 8.0, 100 mM NaCl, and 0.5 mM TCEP. For Fig. 2a, 1 μM of GST-DDX39B or GST was mixed with TREX-2.1 (LENG8$_{490-800}$/PCID2/DSS1) in a binding buffer containing 10 mM Tris pH 8.0, 100 mM NaCl, and 0.5 mM TCEP. For Fig. 8k, each protein was used at 1.5 μM in a binding buffer containing 10 mM Tris pH 8.0, 70 mM NaCl, and 0.5 mM TCEP. IAV NP concentration was calculated based on the trimeric form. For pull-down reactions in Fig. 8k with both TREX-2$^M$/TREX-2.1 and IAV NP, GST-DDX39B was first incubated with TREX-2$^M$ (GANP$_{358-1000}$/PCID2/DSS1) or TREX-2.1 (LENG8$_{490-800}$/PCID2/DSS1) for 10 min on ice, then IAV NP was added, and the mixture was incubated for another 20 min. For all experiments, samples were incubated with Glutathione Sepharose 4B resin on ice for 30 min, with gentle tapping every 3–5 min. Resins were washed with 600 μL binding buffer three times. Proteins remaining on the resin were eluted with buffer containing 10 mM Tris pH 8.0, 100 mM NaCl, 0.5 mM TCEP, and 20 mM reduced glutathione and resolved on SDS-PAGE gel and stained with Instant Blue.

## Reporting summary

Further information on research design is available in the Nature Portfolio Reporting Summary linked to this article.

# Data availability

The cryo-EM maps generated in this study have been deposited in EMDB under accession codes EMD-46981 for TREX-2$^M$/DDX39B, EMD-

46982 for TREX-2$^M$/DDX39B$^{NTM+RecA1}$, EMD-47126 for TREX-2.1/DDX39B, EMD-46985 for TREX-2.1/DDX39B$^{NTM+RecA1}$, and EMD-46983 for TREX-2.1/DDX39B$^{NTM-N}$. The coordinates have been deposited in PDB under accession codes 9DLP for TREX-2$^M$/DDX39B$^{NTM+RecA1}$, 9DLV for TREX-2.1/DDX39B$^{NTM+RecA1}$, and 9DLR for TREX-2.1/DDX39B$^{NTM-N}$. The RNAseq data have been deposited in the SRA BioProject database under accession code PRJNA1240595. Source data are provided with this paper.

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

## Acknowledgements

We thank Melissa Chambers, Scott Collier, and Mariam Haider at the Center for Structural Biology Cryo-EM Facility at Vanderbilt University for assistance in Cryo-EM data collection. We utilized the Glacios cryo-TEM for screening, which was acquired by the NIH S10 award OD030292-01. We thank the Ascano lab for assistance with UV crosslinking experiments. This work was supported by: NIH R35 GM133743 (Y.R.), R01 AI184975 (Y.R. and B.M.A.F.), R01 AI154635 (B.M.A.F) and R01 AI125524 (B.M.A.F.). B.P.C. and A.E.A. were in part supported by NIH/NCI training grant T32CA119925.

## Author contributions

B.P.C., S.G., M.M., D.X., A.E.A., A.V., P.S.H., T.C., Y.X., and Y.R. performed experiments; K.B. and D.X. performed bioinformatics analysis; B.P.C., S.G., M.M., D.X., A.E.A., A.V., P.S.H., T.C., K.B., J.W.S., Y.X., B.M.A.F., and Y.R analyzed the results; B.P.C., S.G., D.X., Y.X., B.M.A.F., and Y.R. wrote the paper.

## Competing interests

The authors declare no competing interests.
