## [Transparent Peer Review file · Nature Communications]

Structural mechanism of DDX39B regulation by human TREX-2 and a related complex in mRNP remodeling

Corresponding Author: Dr Yi Ren

Version 0:

Reviewer comments:

Reviewer #1

(Remarks to the Author)

Clarke et al have identified a new component of the nuclear mRNA processing machinery, the TREX-2.1 complex that, together with the TREX-2 complex, contributes to remodeling of mRNPs by the DEAD-box ATPase, DDX39B (UAP56). The TREX-2.1 complex is based on a core of LENG8 that has a structure similar to part of GAMP that forms the core of the TREX-2 complex. Cryo-EM structures indicate that both LENG8 and GAMP have a distinctive “trigger” loop in their PCI domain that contributes to regulating DDX39B by increasing its ATPase activity. Overall, the structural work has been performed to a high standard and is supported by a series of complementary experiments. However, although this work provides new insights into the structural and functional roles of the TREX-2 and TREX-2.1 complexes in mRNP processing (by regulating DDX39B) and will be of considerable general interest, especially to the mRNA processing and nuclear export communities, it does not, as the authors claim, “lead to a mechanistic understanding of the full functional cycle of DDX39B in mRNP processing” because the way in which DDX39B recognizes and modifies RNA remains unclear. Consequently, it would be helpful if the authors could modify this aspect of the manuscript and also the title to avoid confusing non-specialist readers.

1. Although initially called “helicases”, DEAD-box enzymes are now more commonly called “ATPases” and it would probably be helpful to use this terminology throughout the manuscript (eg lines 145, 148, etc). Also, although DDX39B is now the preferred name for this ATPase, so much of the earlier literature refers to it as UAP56 and so I think it would be helpful to note this in the Abstract and early in the Introduction.
2. It is thought that the two RecA domains of DEAD-box ATPases bind RNA in a way that is incompatible with it retaining a local duplex (double helical) structure (eg ref 109) and so binding results in local remodeling of RNA structure (such as stem loops) and/or protein binding/release, after which ATP hydrolysis leads to the release of the DEAD-box ATPase. Although the authors have identified a key trigger loop in both GAMP and LENG8 that enhances its ATPase activity and dissociation from RNA, it is not clear precisely how DDX39B binds to mRNA in vivo (the only structural work appears to be based on in vitro binding to U15) and the extent to which the binding of TREX-2 or TREX2.1 (or indeed other accessory proteins, such as ALYREF) to mRNA may influence this binding. Nor is it clear precisely how the local mRNP structure and/or the proteins bound to it is altered by the DDX39B cycle. Although, of course, it would not be necessary for the authors to provide complete answers to these questions, it would be helpful if they could discuss them and in particular tone down their claims that their work, excellent as it is, provides a complete understanding of the function of DDX39B in mRNP remodeling. Perhaps something like “adds to” might be more appropriate.
3. I think it would be helpful to revise the title of the manuscript. The remodeling of the mRNP is done by DDX39B – TREX-2 and TREX-2.1 may contribute to and/or modulate/enhance this.
4. The Laskey lab (ref.73) indicated that TREX-2 appeared to be more important for transcripts that had fewer introns. I wondered whether perhaps TREX-2.1 might tend to complement TREX-2 and have a preference for transcripts with larger numbers of introns. These authors also proposed a chaperonin-like function for TREX-2 (perhaps in addition to assisting mRNP remodelling) and I wondered whether TREX-2.1 might also have a similar role as well as its function with DDX39B.
5. I think it would be helpful to say a little more about the composition of TREX-2.1 in the Abstract – at least mention LENG8, the PCI fold, PCID2, DSS1, etc. I hope the Editor would relax any length restrictions to enable this to be done. Also the final

sentence of the Abstract (line 11) could be omitted.

Minor points:

6. "splits" does not seem the best way to describe the changes in DDX39B on line 348 and the legend to Figure 4.

7. line 24 TRanscription-EXport (upper case X).

8. Line 18: Mature mRNPs are often thought of as those that have been spliced and polyadenylated but not yet remodeled to form export-competent mRNPs by adding transport factors (NXF1-NXT1) and removing accessory proteins such as ALYREF. Perhaps this paragraph could be extended a little to avoid any confusion.

9. line 78: It would be helpful here to define TREX-2M

10. Adding a simple schematic to Fig 3 showing the different interactions would make it easier to follow the interactions between the trigger loop of GANP and DDX39B/ADP.

11. line 341 "adds to" rather than "completes" understanding

12. It would help to define "IAV"

Reviewer #2

(Remarks to the Author)

In the manuscript Structure-based mechanism of mRNP remodeling and nuclear export by human TREX-2 and a related complex, Clarke and colleagues identify a novel subcomplex of the essential mRNP remodeling TREX-2 complex that they term TREX-2.1 comprising the proteins LENG8/PCID2/DSS1. They demonstrate this subcomplex is localized diffusively through the nucleus unlike canonical TREX-2 that localizes to the nuclear periphery. Furthermore, they demonstrate that TREX-2.1 interacts with the ATP dependent RNA helicase DDX39B and perform structural analysis via cryo-EM single particle analysis. Finally, they demonstrate, via knockdown of the TREX-2.1 component LENG8, that TREX-2 and TREX-2.1 remodel different subsets of mRNAs.

These results are impressive, and will be of great interest to the community at large; this will make an excellent addition to the journal. However, before publication of this manuscript, the authors should address some major and minor concerns, including potential artifacts in the cryo-EM reconstructions, that would lead to a reduced impact of this work.

Major issues:

1. The cryo-EM reconstruction of TREX- 2M/DDX39BNTM+RecA1 (EMD-46982) is a beautiful, isotropic map and the model derived from this map correlates well. However, the other reconstructions in this manuscript appear to suffer from a preferential orientation and, as such, may be somewhat anisotropic. The sphericity of these reconstructions is below 90%, which merits additional consideration to assess whether the reconstructions are problematic. Unfortunately, the data need to assess this is not present as the authors use the Viewing Direction Distribution plot from cryoSPARC instead of the appropriate Posterior Precision Directional Distribution plot. This is discussed at length in the cryoSPARC user guide: <https://guide.cryosparc.com/processing-data/tutorials-and-case-studies/tutorial-common-cryosparc-plots>. However, Viewing Direction Distribution generally cannot be directly used to infer if the dataset has preferred orientation issues, because the viewing direction distribution doesn't directly visualize the directions along which the volume is well-sampled. The orientation diagnostics job provides a more thorough set of tools for diagnosing orientation issues, as well as in the community discussion: <https://discuss.cryosparc.com/t/posterior-precision-plot-meaning/6534>. The authors should consider replacing the Viewing Direction Distribution plots with the more appropriate Posterior Precision Directional Distribution plot. Alternatively, new versions of cryoSPARC have a "Orientation Diagnostic" job that has a more robust set of Fourier completeness metrics. The authors should strongly consider running that job and providing the output to ensure there is no anisotropy in those maps.

2. The authors should consider using locally filtered or low pass filtered maps for many of the maps where the resolution is not the most important finding of the structure. For example, in Fig. 6a and the associated map, the localization of the two RecA domains is the important feature. This can be appreciated even with a 5 angstrom, low pass filtered map and would increase the power of the fit, since the relative motion of those domains is likely what is causing the poor correlation at higher spatial frequencies. As presented in Fig. 6a, this map does not appear to be a 3.5 angstrom map, as no secondary structure is visible. To bring out the lower resolution portions of the map, the authors should therefore consider presenting a locally filtered map or lowpass filter the map to the point where the two recA domains can be seen at a reasonable threshold.

3. It is unclear whether DSS1 is detected in the GST-LENG8 pulldown, or is this being inferred? The text suggests the entire PCID2 subcomplex is isolated, but DSS1 isn't visible in 1b.

4. Is there any interpretation of the speckle pattern of LENG8 localization in 1d? How do the other components of the suggested complexes localize? Given the suggested hypothesis that the differential localization of TREX-2.1 vs TREX-2 is relevant to its function, it would be useful to explain these localization patterns more thoroughly.

5. The effects of LENG8 silencing on mRNA export could use a bit more context to compare with TREX-2. For example, are the effects on these transcripts more or less severe than seen with GANP? Is there an overall detectable mRNA export defect, and how does this compare to GANP?

Minor points:

1. TREX-2M is introduced as an abbreviation (line 78) without explanation. Prior reference to TREX-2 is just denoted as TREX-2. There should be clarification of TREX-2M.
2. At line 361 there should be a citation for LENG8 interactions.
3. It does not appear as if the secondary structure for LENG8 in the file 9DLR_TREX-2.1/DDX39B(NTM-N).pdb is defined. The authors may want to verify this as it may not display well across programs.

Reviewer #3

(Remarks to the Author)

Nuclear processing and export of mRNPs are critical to proper gene expression and the conserved DEAD-box helicase DDX39B plays important roles in these processes. TREX2 is a nuclear basket associated complex known to interact with export factors to facilitate mRNP translocation through NPCs. To date, the precise mechanisms regulating mRNP remodeling and DDX39B removal at NPCs is unclear. In this manuscript, Clarke et. al use a combination of structural biology and biochemistry, with some supporting in cell experiments, to investigate the relationship between TREX2 and DDX39B. Additionally, they identify an alternative complex named TREX2.1, containing LENG8 in place of GANP, which they also functionally characterize. They demonstrate capability of both TREX and TREX2.1 to form a complex with, and stimulate RNA release from, DDX39B. However, it is not entirely clear from experiments whether hydrolysis of ATP is required for this release. Cryo-EM structures of both TREX and TREX2.1 in complex with DDX39B were determined, showing DDX39B in a similarly open conformation in both structures, contrasted with conformation taken when clamped on RNA. From both structures, a conserved “trigger loop” is identified which inserts into the binding pocket of DDX39B. Mutant experiments demonstrate necessity of these loops for stimulating ATPase activity and RNA release from DDX39B. Finally, they characterize impact of LENG8 knockdown on export status of eight candidate transcripts and demonstrate that the viral protein IAV NP can disrupt association of DDX39B with TREX2/2.1. Without expertise in Cryo-EM data collection, processing, and analysis, this aspect of the manuscript is not commented on in this review. Overall, the manuscript provides valuable data to the field, both detailing and expanding knowledge of DDX39B regulation.

Suggestions for both improved experimental rigor and clarity of figures/text are as follows:

Major Points:

1. Addressing whether ATP hydrolysis is required for RNA release. With the ATP-gamma-S unloading assays (Figs. 2B, 2C, 7D), it is unknown whether ATP hydrolysis is required for RNA release, since some DEAD-box helicases can hydrolyze ATP-gamma-S. Two potential activities being hydrolysis with release of the RNA or unloading of DDX39B without hydrolysis. To clarify this mechanistic difference, it is recommended the RNA release assay is repeated with an ATPase dead mutant of DDX39B or with ADP-BeFx.
2. Clarity in comparison of mutant and wild-type activity. For assays measuring ATPase activity (2D and 7C) wild-type need to be included alongside mutants for comparison (as is done in fig 4E). If WT was not included when mutant experiments were run, it is even more important to repeat including WT for rigor and reproducibility in making comparison of activity head-to-head. This likewise applies to unloading assays, all conditions should be on the same gel (combine 2B & 2C, inclusion of WT in 7D).
3. Addition of functional experiments to support roles of TREX2 and TREX2.1. In figure 8, siGANP should be included if the authors want to make argument of “GANP-dependency” and overlapping vs. distinct targets. These transcripts come from data using an entirely different parental cell line (DLD-1) and were identified in the context of GANP auxin-induced degradation, which yielded higher knockdown than observed here with siLENG8. Furthermore, they only included targets which did not show a change in overall expression by RNA-seq, which again was not investigated here. The authors should validate whether these transcripts show the same dependencies using siGANP or AID-GANP in context of experimental parameters used here if want to be able to make a strong point about overlap of targets. Alternatively, the authors can moderate language to acknowledge the different cell lines and knockdowns with regards to the overall premise of GANP-dependence vs. independence.

In this same vein, immunofluorescence in figure 1D would be strengthened by additional co-staining to examine relationships described in text. Some degree of colocalization with PCID2 and DSS1 is expected if making the point these form a trimeric complex in vivo (or an IP to show PCID2 and DSS1 pull down with LENG8 from cells if antibodies for staining are not readily available). In relation to discussion of nuclear speckles starting at line 354, it would again strengthen data to co-stain for a reliable marker of nuclear speckles with LENG8 and/or include DDX39B staining as well if want to make point that LENG8 is regulating DDX39B in the nucleus.

Finally, the conclusion that TREX2.1 has a role in mRNP export (line 315) is not strongly supported by the presented data. For example, changes in RNA surveillance, splicing, or other mRNA processing events could cause nuclear accumulation

of mRNA indirectly related to mRNA export. It may be informative to examine nuclear pA levels by dT FISH in the context of the siLENG8 experiments; however, significant work will be required to determine if TREX2.1 is directly involved in nuclear export of mRNPs.

Overall, without these functional experiments, the language should be amended to make clear what is speculative based on current data. While biochemical and structural data are very strong, the current discussion of functional roles of TREX2 vs TREX2.1 in vivo are not warranted by the data in the manuscript.

Other Points:

4. Additional clarity would benefit the reader when discussing TREX in the introduction. Some parts of introduction are unclear in terms of which organism is being discussed and effort should be paid in making this clearer when discussing both mammalian and yeast counterparts. For example, making clear data referenced in line 23-24 comes only from mammalian structural work. Furthermore, phrasing comes across as though TREX components are always in complex and accomplish all their functions as part of this complex, when data suggests otherwise. Subunits have very different expression levels, and some are essential while others are not. Subunits are also tied to varying functions, for example Sub2/DDX39B is tied to function in splicing while THO is not. So, while they are known to form complex engaging mRNPs, care should be taken to accurately describe association and gaps in knowledge. One example being lines 42-43 which suggests TREX complex association with phospho CTD. Data cited more accurately shows association of certain TREX components and there is not direct data supporting recruitment of the assembled complex. Line 27-28 also assumes temporality of DDX39B/Sub2 being primed by THO prior to loading in mRNP, which is not directly supported by cryo-EM structure work cited.

5. The authors can consider referencing and/or discussion of a pre-print coming to similar conclusions (<https://doi.org/10.1101/2024.03.24.586400>). While not yet published in peer-reviewed journal, work drawing very similar conclusions has been on BioRxiv since March 2024. While it is ultimately up to the authors whether to include discussion of a pre-print, it strongly supports the authors' own conclusions and would strengthen the current manuscript. Furthermore, it would be in support of open and transparent science and benefit cohesive knowledge of the field at large.

6. Given use in the literature, it should be noted in the introduction that DDX39B is also commonly referred to as UAP56 for clarity of readers.

7. Fig 1A is hard to read clearly, for example ENY2 and Centrin are not immediately clear. This figure panel would benefit from showing all proteins in the same linear domain format. The panel could also include a structural model/cartoon of known interactions.

8. It is hard to see the overlay in Fig 4A. It would be helpful to either just show RecA lobes without other subunits or make the other subunits more transparent to aid in seeing RecA conformations clearly.

9. Figure 8D is not very informative and implies certain localized activity (both of TREX2.1 and IAV NP) that is not directly supported by the data in the manuscript. The panel also implies info on TREX2 / TREX 2.1 binding in a 3' end biased fashion. The mRNA is also portrayed without a cap or pA tail and as a linear molecule. This is not ideal when portraying models of mRNPs with protein complexes important to compaction and packaging of mRNPs. It would be better to just incorporate bars suggesting inhibition of these complexes by IAV NP into 8C and make this panel larger.

Reviewer #4

(Remarks to the Author)

Reviewer #5

(Remarks to the Author)

Version 1:

Reviewer comments:

Reviewer #1

(Remarks to the Author)

The authors have addressed all of my criticisms and suggestions satisfactorily. Moreover, as well as improving the processing of the cryoEM data for TREX2.1 and its complex, they have expanded the information available on the

complexes formed between TREX-2 and DDX38b (UAP56) that provides important information on how export-competent mRNPs are generated and the text has been expanded to discuss this in greater detail. They have also demonstrated that the TREX-2 and TREX2.1 complexes show different preferences for the mRNAs for which they contribute to generating export-competent mRNPs. Overall this is now a very nice piece of work that will be of great interest in the field.

Reviewer #2

(Remarks to the Author)

The authors have addressed essentially all our points satisfactorily. We note, the relevance of the transcript comparisons between the LENG8 and GANP knockdowns would be much clearer if these were done in parallel in the same conditions, or at least had additional controls or explanations demonstrating their equivalence. There seem to be many variables between the two datasets that could contribute to the differences seen – different cell line used, knockdown efficiency, and other potential differences in experimental details, e.g. method and efficiency of nuclear/cytoplasmic fractionation. However, this is a point that another review had a strong focus on, so we're happy to leave this to them and the editor. Otherwise, this is overall a beautiful piece of work.

Reviewer #3

(Remarks to the Author)

Overall, Clarke et. al addressed reviewer comments thoroughly and have improved the quality of an already rigorous manuscript. The EMSA experiments with AMPPNP aid in clarifying that ATP hydrolysis is not required for DDX39B unloading and the additional experiments directly comparing ATPase and unloading activities strengthens the conclusions drawn. Edits to text and figures for improved background and clarity based on suggestions are appreciated. The primary issues remaining relate to in cell experiments addressed in original Major Point #3. While RNA sequencing and additional immunofluorescence experiments were performed to address these points, concerns remain regarding implementation and interpretation. These can be addressed by amending figure display and text, not requiring additional experiments. The concerns remaining are:

1. Direct comparison of “GANP-dependent” and “LENG8-dependent” RNA features in figure 8.

The original concern regarding the argument of overlapping vs distinct targets between GANP and LENG8 still stands. While the RNA sequencing with siLENG8 is valuable and provides great insight into changes in gene expression upon LENG8 depletion, direct comparison to previously published siGANP data seems to be inappropriate. This data comes from a different cell line (HCT116 vs A549 here) and as such there likely are differences in these transcriptomes which could influence the impacts of depletions. Furthermore, nuclear vs cytoplasmic isolation was performed differently, and gene expression analysis was performed by microarray which has significantly less depth than RNA sequencing done here. For these reasons, it is suggested that Figure 8 is adapted to not include the “GANP-dependent” plots. If wanting to make a comparison of “LENG8-dependent” features, it would be more appropriate to compare to features of the overall transcriptome in untreated A549 control cells. In order to make a strong point about shared vs distinct targets of GANP and LENG8, depletion and gene expression analyses would need to be done with the same experimental conditions. Text in paragraphs starting at lines 349 and 437 should be amended according to any changes made.

2. Evidence of LENG8 function within nuclear speckles is weak.

Immunofluorescence for SC35 to corroborate LENG8 colocalization with nuclear speckles is appreciated. However, overall colocalization is not all that strong and a large majority of LENG8 is localized outside of SC35 signal. Furthermore, there is still no staining for DDX39B. While a role for DDX39B in nuclear speckles is well documented, there is no evidence that DDX39B and LENG8 co-occupy the same speckles and may be residing in distinct pools of speckles.

3. More precise language when describing role of TREX2.1 in nuclear export.

Even with added RNA sequencing, experiments still do not strongly support a direct role for TREX2.1 in mRNA export. Authors note “...we selected mRNAs whose total cell levels are not altered but are retained in the nucleus upon LENG8 knockdown. In this manner, we selected the mRNAs whose nucleocytoplasmic transport is affected, excluding other effects on mRNA levels.” However, while this does determine transcripts with impacted N/C ratios in the event of LENG8 knockdown, it does not prove this is due to direct disruption in export, nor exclude the possibility this stems from defects in other processes. For example, decreased nuclear surveillance would also lead to higher N/C ratios, although not a defect in export mechanisms per se. Furthermore, as cited on line 421, LENG8 does associate with quality control related factors, making this a distinct possibility for the changed N/C ratios observed in the sequencing data. With this in mind, the title on line 327 should be edited to read “influences nucleocytoplasmic ratio” rather than “mediates nuclear export” and edits should be made in this section to not overstate the role, as well as in the discussion starting at line 437. While GANP was demonstrated to play a direct role in export, many of the same experiments leading to this conclusion were not repeated here, and so LENG8 cannot be assumed to play this same role without proper supporting data.

Reviewer #4

(Remarks to the Author)

Reviewer #5

(Remarks to the Author)

Point-by-point response to reviewer comments

We thank all the Reviewers for their constructive comments. We have revised the manuscript and below is a point-by-point response to the comments.

Reviewer #1:

Clarke et al have identified a new component of the nuclear mRNA processing machinery, the TREX-2.1 complex that, together with the TREX-2 complex, contributes to remodeling of mRNPs by the DEAD-box ATPase, DDX39B (UAP56). The TREX-2.1 complex is based on a core of LENG8 that has a structure similar to part of GAMP that forms the core of the TREX-2 complex. Cryo-EM structures indicate that both LENG8 and GANP have a distinctive “trigger’ loop in their PCI domain that contributes to regulating DDX39B by increasing its ATPase activity. Overall, the structural work has been performed to a high standard and is supported by a series of complementary experiments. However, although this work provides new insights into the structural and functional roles of the TREX-2 and TREX-2.1 complexes in mRNP processing (by regulating DDX39B) and will be of considerable general interest, especially to the mRNA processing and nuclear export communities, it does not, as the authors claim, “lead to a mechanistic understanding of the full functional cycle of DDX39B in mRNP processing” because the way in which DDX39B recognizes and modifies RNA remains unclear. Consequently, it would be helpful if the authors could modify this aspect of the manuscript and also the title to avoid confusing non-specialist readers.

We thank the reviewer for her/his very helpful suggestions, and we address her/his comments in detail below.

1. Although initially called “helicases”, DEAD-box enzymes are now more commonly called “ATPases” and it would probably be helpful to use this terminology throughout the manuscript (eg lines 145, 148, etc). Also, although DDX39B is now the preferred name for this ATPase, so much of the earlier literature refers to it as UAP56 and so I think it would be helpful to note this in the Abstract and early in the Introduction.

Response: Point accepted. In this revised manuscript, we use “ATPase” to refer to DEAD-box proteins throughout the manuscript. We also add the name “UAP56” in abstract and when DDX39B is first introduced in the introduction.

2. It is thought that the two RecA domains of DEAD-box ATPases bind RNA in a way that is incompatible with it retaining a local duplex (double helical) structure (eg ref 109) and so binding results in local remodeling of RNA structure (such as stem loops) and/or protein binding/release, after which ATP hydrolysis leads to the release of the DEAD-box ATPase. Although the authors have identified a key trigger loop in both GAMP and LENG8 that enhances its ATPase activity and dissociation from RNA, it is not clear precisely how DDX39B binds to mRNA in vivo (the only structural work appears to be based on in vitro binding to U15) and the extent to which the binding of TREX-2 or TREX2.1 (or indeed other accessory proteins, such as ALYREF) to mRNA may influence this binding. Nor is it clear precisely how the local mRNP structure and/or the proteins bound to it is altered by the DDX39B cycle. Although, of course, it would not be necessary for the authors to provide complete answers to these questions, it would be helpful if they could discuss them and in particular tone down their claims that their work, excellent as it is, provides a complete

understanding of the function of DDX39B in mRNP remodeling. Perhaps something like "adds to" might be more appropriate.

Response: We have changed "complete" to "add to" in the revised manuscript (line 386). We also add discussion on mRNP remodeling in vivo (line 433-435). We agree with the reviewer that the mRNA substrates and the remodeling activity of DDX39B/Sub2 in cells remain unclear. To better understand the RNA substrates of DDX39B and its binding partners, we have obtained new RNAseq data to identify TREX-2.1 (LENG8) dependent mRNAs in this revision (see response to point 4).

3. I think it would be helpful to revise the title of the manuscript. The remodeling of the mRNP is done by DDX39B – TREX-2 and TREX-2.1 may contribute to and/or modulate/enhance this.

Response: Point accepted. The title has been revised to "Structure-based mechanism of DDX39B regulation by human TREX-2 and a related complex in mRNP remodeling and nuclear export".

4. The Laskey lab (ref.73) indicated that TREX-2 appeared to be more important for transcripts that had fewer introns. I wondered whether perhaps TREX-2.1 might tend to complement TREX-2 and have a preference for transcripts with larger numbers of introns. These authors also proposed a chaperonin-like function for TREX-2 (perhaps in addition to assisting mRNP remodelling) and I wondered whether TREX-2.1 might also have a similar role as well as its function with DDX39B.

Response: The reviewer is correct. In fact, we have now performed RNAseq analysis of whole cell lysates, nuclear and cytoplasmic fractions of cells that were subjected to LENG8 knockdown versus cells treated with siRNA control. As shown in Figure 8, we selected mRNAs whose total cell levels are not altered but are retained in the nucleus upon LENG8 knockdown. In this manner, we selected the mRNAs whose nucleocytoplasmic transport is affected, excluding other effects on mRNA levels. We found that TREX-2.1 facilitates the nuclear export of transcripts with more exons (or larger number of introns), shorter exon length, and higher GC content compared to GANP-dependent mRNAs. These LENG8 knockdown results were compared to siRNA knockdown of GANP previously reported (PMID: 24510098).

As pointed out by the Reviewer, TREX-2 is suggested to facilitate mRNP recruitment and handover to the NPC. One of the major differences between TREX-2 and TREX-2.1 is that TREX-2.1 does not seem to contain an apparent nuclear pore complex targeting domain. Consistently, the LENG8 subunit of TREX-2.1, in contrast to the GANP subunit of TREX-2, is not enriched at the NPC (Fig. 1d). It is unclear whether TREX-2.1 has a similar role and would be an interesting question for future investigation.

5. I think it would be helpful to say a little more about the composition of TREX-2.1 in the Abstract – at least mention LENG8, the PCI fold, PCID2, DSS1, etc. I hope the Editor would relax any length restrictions to enable this to be done. Also the final sentence of the Abstract (line 11) could be omitted.

Response: We have revised the abstract in response to the reviewer's comments. The current

abstract includes the composition of TREX-2.1, and the last sentence in the original version has been deleted.

Minor points:

6. “splits” does not seem the best way to describe the changes in DDX39B on line 348 and the legend to Figure 4.

Response: We rephrase it as “inserts between the two RecA domains”.

7. line 24 TRanscription-EXport (upper case X).

Response: This is now corrected.

8. Line 18: Mature mRNPs are often thought of as those that have been splices and polyadenylated but not yet remodeled to form export-competent mRNPs by adding transport factors (NXF1-NXT1) and removing accessory proteins such as ALYREF. Perhaps this paragraph could be extended as little to avoid any confusion.

Response: We have revised this sentence as the following: “Following proper processing of the mRNA and acquisition/removal of specific proteins, export-competent mRNPs travel from the nucleus to the cytoplasm through the nuclear pore complex (NPC).”

9. line 78: It would be helpful here to define TREX-2M

Response: In this revised manuscript, we change TREX-2^M to TREX-2 in the introduction as well as in the abstract for better readability.

10. Adding a simple schematic to Fig 3 showing the different interactions would make it easier to follow the interactions between the trigger loop of GANP and DDX39B/ADP.

Response: We add a schematic (Fig. 3h) as suggested.

11. line 341 “adds to” rather than “completes” understanding

Response: We have made the suggested change (line 386 in the revised manuscript)

12. It would help to define “IAV”

Response: In this revised manuscript, Influenza A Virus (IAV) is defined in line 91.

Reviewer #2:

In the manuscript Structure-based mechanism of mRNP remodeling and nuclear export by human TREX-2 and a related complex, Clarke and colleagues identify a novel subcomplex of the essential mRNP remodeling TREX-2 complex that they term TREX-2.1 comprising the proteins LENG8/PCID2/DSS1. They demonstrate this subcomplex is localized diffusively through the nucleus unlike canonical TREX-2 that localizes to the nuclear periphery. Furthermore, they demonstrate that TREX-2.1 interacts with the ATP dependent RNA helicase DDX39B and perform structural analysis via cryo-EM single particle analysis. Finally, they demonstrate, via knockdown of the TREX-2.1 component LENG8, that TREX-2 and TREX-2.1 remodel different subsets of mRNAs.

These results are impressive, and will be of great interest to the community at large; this will make an excellent addition to the journal. However, before publication of this manuscript, the authors should address some major and minor concerns, including potential artifacts in the cryo-EM reconstructions, that would lead to a reduced impact of this work.

We thank the Reviewer's comments and suggestions. In this revised manuscript, we are excited to present updated cryo-EM maps of TREX-2.1/DDX39B with major improvements in both the orientation distribution and the resolution, as detailed below.

Major issues:

1. The cryo-EM reconstruction of TREX- 2M/DDX39BNTM+RecA1 (EMD-46982) is a beautiful, isotropic map and the model derived from this map correlates well. However, the other reconstructions in this manuscript appear to suffer from a preferential orientation and, as such, may be somewhat anisotropic. The sphericity of these reconstructions is below 90%, which merits additional consideration to assess whether the reconstructions are problematic. Unfortunately, the data need to assess this is not present as the authors use the Viewing Direction Distribution plot from cryoSPARC instead of the appropriate Posterior Precision Directional Distribution plot. This is discussed at length in the cryoSPARC user guide: <https://guide.cryosparc.com/processing-data/tutorials-and-case-studies/tutorial-common-cryosparc-plots>. However, Viewing Direction Distribution generally cannot be directly used to infer if the dataset has preferred orientation issues, because the viewing direction distribution doesn't directly visualize the directions along which the volume is well-sampled. The orientation diagnostics job provides a more thorough set of tools for diagnosing orientation issues, as well as in the community discussion:<https://discuss.cryosparc.com/t/posterior-precision-plot-meaning/6534>. The authors should consider replacing the Viewing Direction Distribution plots with the more appropriate Posterior Precision Directional Distribution plot. Alternatively, new versions of cryoSPARC have a "Orientation Diagnostic" job that has a more robust set of Fourier completeness metrics. The authors should strongly consider running that job and providing the output to ensure there is no anisotropy in those maps.

Response: We thank the Reviewer for the suggestions, which prompted us to revise the cryo-EM reconstruction workflow. Given the modest size of TREX-2.1/DDX39B and the heterogeneity of the sample, one potential cause of orientation bias is poor particle alignment and the orientations that are hard to align were discarded during the early stages of particle sorting. In this revision, we used reference maps obtained from pilot processing for the first step of 3D sorting to separate good particles and junk particles. These reference maps, having undergone multiple rounds of 3D sorting, show better quality than the ab-initio reference maps utilized in the original

submission. This revised workflow, as detailed in Supplementary Fig. 5, has retained particles with better orientation distribution.

Here, we present the cryo-EM reconstruction of TREX-2.1/DDX39B^{NTM-RecA1} at 2.97 Å resolution. This map shows a sphericity of 0.930 as analyzed by 3DFSC (Supplementary Fig. 8d). This reconstruction represents a major improvement compared to the 3.36 Å map with a sphericity of 0.858 in the original submission. A comparison of the TREX-2.1/DDX39B^{NTM-RecA1} maps presented in the original submission and the revised submission (Fig. 6c) is shown below.

Likewise, the cryo-EM reconstruction of TREX-2.1/DDX39B^{NTM-N} is also significantly improved, with an overall resolution at 3.08 Å and a sphericity of 0.952 (Supplementary Fig. 6d). This is in contrast with the original reconstruction at 3.35 Å resolution and a sphericity of 0.811. A comparison of the TREX-2.1/DDX39B^{NTM-N} maps presented in the original submission and the revised submission (Fig. 5a) is shown below.

Additionally, following the Reviewer's comment, Posterior Precision Directional Distribution plots are used to indicate orientation distribution in this revised manuscript.

2. The authors should consider using locally filtered or low pass filtered maps for many of the maps where the resolution is not the most important finding of the structure. For example, in Fig. 6a and the associated map, the localization of the two RecA domains is the important feature.

This can be appreciated even with a 5 angstrom, low pass filtered map and would increase the power of the fit, since the relative motion of those domains is likely what is causing the poor correlation at higher spatial frequencies. As presented in Fig. 6a, this map does not appear to be a 3.5 angstrom map, as no secondary structure is visible. To bring out the lower resolution portions of the map, the authors should therefore consider presenting a locally filtered map or lowpass filter the map to the point where the two recA domains can be seen at a reasonable threshold.

Response: We thank the reviewer for the comment. Fig. 3b (TREX-2/DDX39B) and Fig. 6a (TREX-2.1/DDX39B) are now presented with low-pass filtered maps. Because the RecA2 domain of DDX39B is highly dynamic and shows weaker density compared to other parts of the complexes, low contour levels have to be used, so the secondary structures are not evident. It is worth mentioning that the updated cryo-EM data processing workflow, as detailed in response to point 1, also leads to improvement of the RecA2 densities in the TREX-2.1/DDX39B map.

3. It is unclear whether DSS1 is detected in the GST-LENG8 pulldown, or is this being inferred? The text suggests the entire PCID2 subcomplex is isolated, but DSS1 isn't visible in 1b.

Response: We repeated the GST-LENG8 pull down experiment and loaded more sample on the gel for better visualization of the small DSS1 protein (70 residues). The new experiment is shown in Supplementary Fig. 1a.

4. Is there any interpretation of the speckle pattern of LENG8 localization in 1d? How do the other components of the suggested complexes localize? Given the suggested hypothesis that the differential localization of TREX-2.1 vs TREX-2 is relevant to its function, it would be useful to explain these localization patterns more thoroughly.

Response: Regarding speckle pattern, we have co-stained LENG8 with a nuclear speckle marker and show that a subset of LENG8 colocalizes with nuclear speckles (Supplementary Fig. 1b). We also performed immunoprecipitation of LENG8 with PCID2 and DDX39B (UAP56) to show the interaction between these proteins in human cells (Fig. 2b). Regarding DDX39B (UAP56), it has been shown that a subset of DDX39B (UAP56) is localized at nuclear speckles (PMID: 30194269). Thus, a pool of both LENG8 and DDX39B (UAP56) is found at nuclear speckles.

5. The effects of LENG8 silencing on mRNA export could use a bit more context to compare with TREX-2. For example, are the effects on these transcripts more or less severe than seen with GANP? Is there an overall detectable mRNA export defect, and how does this compare to GANP?

Response: We have now performed RNAseq analysis of whole cell lysates, nuclear and cytoplasmic fractions of cells that were subjected to LENG8 knockdown versus cells treated with siRNA control. As shown in Figure 8, we selected mRNAs whose total cell levels are not altered but are retained in the nucleus upon LENG8 knockdown. In this manner, we selected the mRNAs whose nucleocytoplasmic transport is affected, excluding other effects on mRNA levels. We found that TREX-2.1 facilitates the nuclear export of transcripts with more exons (or larger number of introns), shorter exon length, and higher GC content compared to GANP-dependent mRNAs. These LENG8 knockdown results were compared to siRNA knockdown of GANP previously reported (PMID: 24510098).

Minor points:

1. TREX-2M is introduced as an abbreviation (line 78) without explanation. Prior reference to TREX-2 is just denoted as TREX-2. There should be clarification of TREX-2M.

Response: In this revised manuscript, TREX-2^M is replaced with TREX-2 in the introduction. TREX-2^M is introduced later in the cryo-EM studies.

2. At line 361 there should be a citation for LENG8 interactions.

Response: Citations have been added (line 419 and 421 in this revision).

3. It does not appear as if the secondary structure for LENG8 in the file 9DLR_TREX-2.1/DDX39B(NTM-N).pdb is defined. The authors may want to verify this as it may not display well across programs.

Response: The coordinates associated with this submission contain secondary structure definition.

Reviewer #3:

Nuclear processing and export of mRNPs are critical to proper gene expression and the conserved DEAD-box helicase DDX39B plays important roles in these processes. TREX2 is a nuclear basket associated complex known to interact with export factors to facilitate mRNP translocation through NPCs. To date, the precise mechanisms regulating mRNP remodeling and DDX39B removal at NPCs is unclear. In this manuscript, Clarke et. al use a combination of structural biology and biochemistry, with some supporting in cell experiments, to investigate the relationship between TREX2 and DDX39B. Additionally, they identify an alternative complex named TREX2.1, containing LENG8 in place of GANP, which they also functionally characterize. They demonstrate capability of both TREX and TREX2.1 to form a complex with, and stimulate RNA release from, DDX39B. However, it is not entirely clear from experiments whether hydrolysis of ATP is required for this release. Cryo-EM structures of both TREX and TREX2.1 in complex with DDX39B were determined, showing DDX39B in a similarly open conformation in both structures, contrasted with conformation taken when clamped on RNA. From both structures, a conserved “trigger loop” is identified which inserts into the binding pocket of DDX39B. Mutant experiments demonstrate necessity of these loops for stimulating ATPase activity and RNA release from DDX39B. Finally, they characterize impact of LENG8 knockdown on export status of eight candidate transcripts and demonstrate that the viral protein IAV NP can disrupt association of DDX39B with TREX2/2.1. Without expertise in Cryo-EM data collection, processing, and analysis, this aspect of the manuscript is not commented on in this review. Overall, the manuscript provides valuable data to the field, both detailing and expanding knowledge of DDX39B regulation.

Suggestions for both improved experimental rigor and clarity of figures/text are as follows:

We thank the Reviewer for all the helpful comments. We have performed a series of experiments as suggested to strengthen the manuscript.

Major Points:

1. Addressing whether ATP hydrolysis is required for RNA release. With the ATP-gamma-S unloading assays (Figs. 2B, 2C, 7D), it is unknown whether ATP hydrolysis is required for RNA release, since some DEAD-box helicases can hydrolyze ATP-gamma-S. Two potential activities being hydrolysis with release of the RNA or unloading of DDX39B without hydrolysis. To clarify this mechanistic difference, it is recommended the RNA release assay is repeated with an ATPase dead mutant of DDX39B or with ADP-BeFx.

Response: Based on the Reviewer's comment, we performed the EMSA assay with the non-hydrolyzable ATP analog, AMPPNP. Our data, included as Fig. 2d, show that TREX-2.1 mediated DDX39B unloading from the preassembled DDX39B/RNA complex does not require ATP hydrolysis.

2. Clarity in comparison of mutant and wild-type activity. For assays measuring ATPase activity (2D and 7C) wild-type need to be included alongside mutants for comparison (as is done in fig 4E). If WT was not included when mutant experiments were run, it is even more important to repeat including WT for rigor and reproducibility in making comparison of activity head-to-head. This likewise applies to unloading assays, all conditions should be on the same gel (combine 2B & 2C, inclusion of WT in 7D).

Response: We have performed additional experiments to directly compare the activities of TREX-2.1 WT and mutant. The ATPase activity was compared in Fig. 7c, and the DDX39B unloading activity was compared in Fig. 7d. Additionally, we have performed further experiments to directly compare the DDX39B unloading activity of the TREX-2.1 complex (previous Fig. 2b) and individual components (previous Fig. 2c) in Fig. 2c in the revised manuscript.

3. Addition of functional experiments to support roles of TREX2 and TREX2.1. In figure 8, siGANP should be included if the authors want to make argument of "GANP-dependency" and overlapping vs. distinct targets. These transcripts come from data using an entirely different parental cell line (DLD-1) and were identified in the context of GANP auxin-induced degradation, which yielded higher knockdown than observed here with siLENG8. Furthermore, they only included targets which did not show a change in overall expression by RNA-seq, which again was not investigated here. The authors should validate whether these transcripts show the same dependencies using siGANP or AID-GANP in context of experimental parameters used here if want to be able to make a strong point about overlap of targets. Alternatively, the authors can moderate language to acknowledge the different cell lines and knockdowns with regards to the overall premise of GANP-dependence vs. independence.

Response: We have now performed RNAseq analysis of whole cell lysates, nuclear and cytoplasmic fractions of cells that were subjected to LENG8 knockdown versus cells treated with siRNA control. As shown in Figure 8, we selected mRNAs whose total cell levels are not altered but are retained in the nucleus upon LENG8 knockdown. In this manner, we selected the mRNAs whose nucleocytoplasmic transport is affected, excluding other effects on mRNA levels. We found that TREX-2.1 facilitates the nuclear export of transcripts with more exons (or larger number of

introns), shorter exon length, and higher GC content compared to GANP-dependent mRNAs. These LENG8 knockdown results were compared to siRNA knockdown of GANP previously reported (PMID: 24510098).

In this same vein, immunofluorescence in figure 1D would be strengthened by additional co-staining to examine relationships described in text. Some degree of colocalization with PCID2 and DSS1 is expected if making the point these form a trimeric complex in vivo (or an IP to show PCID2 and DSS1 pull down with LENG8 from cells if antibodies for staining are not readily available). In relation to discussion of nuclear speckles starting at line 354, it would again strengthen data to co-stain for a reliable marker of nuclear speckles with LENG8 and/or include DDX39B staining as well if want to make point that LENG8 is regulating DDX39B in the nucleus.

Response: As requested, we performed immunoprecipitation of LENG8 with PCID2 and DDX39B (UAP56) to show the interaction between these proteins in human cells (Fig. 2b). We have also co-stained LENG8 with a nuclear speckle marker and show that a subset of LENG8 colocalizes with nuclear speckles (Supplementary Fig. 1b). Regarding DDX39B (UAP56), it has been shown that a subset of DDX39B (UAP56) is localized at nuclear speckles (PMID: 30194269). Thus, a pool of both LENG8 and DDX39B (UAP56) is localized at nuclear speckles.

Finally, the conclusion that TREX2.1 has a role in mRNP export (line 315) is not strongly supported by the presented data. For example, changes in RNA surveillance, splicing, or other mRNA processing events could cause nuclear accumulation of mRNA indirectly related to mRNA export. It may be informative to examine nuclear pA levels by dT FISH in the context of the siLENG8 experiments; however, significant work will be required to determine if TREX2.1 is directly involved in nuclear export of mRNPs.

Overall, without these functional experiments, the language should be amended to make clear what is speculative based on current data. While biochemical and structural data are very strong, the current discussion of functional roles of TREX2 vs TREX2.1 in vivo are not warranted by the data in the manuscript.

Response: As mentioned above, we have now performed RNAseq analysis of whole cell lysates, nuclear and cytoplasmic fractions of cells that were subjected to LENG8 knockdown versus cells treated with siRNA control. As shown in Figure 8, we selected mRNAs whose total cell levels are not altered but are retained in the nucleus upon LENG8 knockdown. In this manner, we selected the mRNAs whose nucleocytoplasmic transport is affected, excluding other effects on mRNA levels. We found that TREX-2.1 facilitates the nuclear export of transcripts with more exon numbers (or larger number of introns), shorter exon length, and higher GC content compared to GANP-dependent mRNAs. This LENG8 knockdown results were compared to siRNA knockdown of GANP previously reported (PMID: 24510098).

Other Points:

4. Additional clarity would benefit the reader when discussing TREX in the introduction. Some parts of introduction are unclear in terms of which organism is being discussed and effort should be paid in making this clearer when discussing both mammalian and yeast counterparts. For example, making clear data referenced in line 23-24 comes only from mammalian structural work. Furthermore, phrasing comes across as though TREX components are always in complex and

accomplish all their functions as part of this complex, when data suggests otherwise. Subunits have very different expression levels, and some are essential while others are not. Subunits are also tied to varying functions, for example Sub2/DDX39B is tied to function in splicing while THO is not. So, while they are known to form complex engaging mRNPs, care should be taken to accurately describe association and gaps in knowledge. One example being lines 42-43 which suggests TREX complex association with phospho CTD. Data cited more accurately shows association of certain TREX components and there is not direct data supporting recruitment of the assembled complex. Line 27-28 also assumes temporality of DDX39B/Sub2 being primed by THO prior to loading in mRNP, which is not directly supported by cryo-EM structure work cited.

Response: We thank the reviewer for the comments and have revised accordingly. We clarify that the CBC-ALYREF interaction refers to studies in mammalian cells (now line 29-31, line 23-24 in the original manuscript).

We separately describe the functions of the yeast and human RNA Pol II interaction with the THO complex in mRNP assembly (now line 49-52, line 42-43 in the original manuscript).

We have also revised line 36-38 (line 27-28 in the original manuscript) as the following: “THO directly binds to DDX39B/Sub2 and positions its two RecA domains in a half-open conformation, as first revealed by a crystal structure of the yeast THO/Sub2 complex.”

5. The authors can consider referencing and/or discussion of a pre-print coming to similar conclusions (<https://doi.org/10.1101/2024.03.24.586400>). While not yet published in peer-reviewed journal, work drawing very similar conclusions has been on BioRxiv since March 2024. While it is ultimately up to the authors whether to include discussion of a pre-print, it strongly supports the authors' own conclusions and would strengthen the current manuscript. Furthermore, it would be in support of open and transparent science and benefit cohesive knowledge of the field at large.

Response: We now include the discussion of the pre-print in the first paragraph of the discussion section, as detailed below.

During the revision of this manuscript, structures of yeast TREX-2 in association with Sub2 were published. Compared to the structures of TREX-2/DDX39B and TREX-2.1/DDX39B, the trigger loop within the yeast TREX-2/Sub2 complex adopts a conformation in between, in which the ADP has been released and the trigger loop is still engaged with Sub2 (Supplementary Fig. 11). A preprint reported an EM reconstruction of TREX-2/DDX39B from a PCID2-DDX39B fusion protein in the presence of AMPPNP and RNA. Consistent with our RNP remodeling assay (Fig. 2d) showing that TREX-2.1 disassembles the RNP composed of AMPPNP, RNA, and DDX39B, RNA is not present in the resulting structure and likely dissociated from DDX39B after TREX-2 mediated remodeling activity. Together, the series of structural snapshots provide mechanistic insights into the full ATPase cycle of the crucial DEAD-box ATPase DDX39B.

6. Given use in the literature, it should be noted in the introduction that DDX39B is also commonly referred to as UAP56 for clarity of readers.

Response: This information is now included in the abstract and when DDX39B is first introduced in the introduction.

7. Fig 1A is hard to read clearly, for example ENY2 and Centrin are not immediately clear. This figure panel would benefit from showing all proteins in the same linear domain format. The panel could also include a structural model/cartoon of known interactions.

Response: ENY2 and Centrin are omitted from Fig. 1a for better focus on the GANP/PCID2/DSS1 part of the TREX-2 complex. ENY2 and Centrin are described in the figure legend.

8. It is hard to see the overlay in Fig 4A. It would be helpful to either just show RecA lobes without other subunits or make the other subunits more transparent to aid in seeing RecA conformations clearly.

Response: As suggested by the reviewer, we have changed the transparency level for the TREX-2 complex in Fig. 4a for better visualization of DDX39B.

9. Figure 8D is not very informative and implies certain localized activity (both of TREX2.1 and IAV NP) that is not directly supported by the data in the manuscript. The panel also implies info on TREX2 / TREX 2.1 binding in a 3' end biased fashion. The mRNA is also portrayed without a cap or pA tail and as a linear molecule. This is not ideal when portraying models of mRNPs with protein complexes important to compaction and packaging of mRNPs. It would be better to just incorporate bars suggesting inhibition of these complexes by IAV NP into 8C and make this panel larger.

Response: Point accepted. Fig. 8d is removed in the revised manuscript.

Point-by-point response to reviewer comments

We appreciate all the reviewers for taking the time to review our manuscript and for offering constructive comments. Below please find our responses to the remaining issues.

Reviewer #1:

The authors have addressed all of my criticisms and suggestions satisfactorily. Moreover, as well as improving the processing of the cryoEM data for TREX2.1 and its complex, they have expanded the information available on the complexes formed between TREX-2 and DDX38b (UAP56) that provides important information on how export-competent mRNPs are generated and the text has been expanded to discuss this in greater detail. They have also demonstrated that the TREX-2 and TREX2.1 complexes show different preferences for the mRNAs for which they contribute to generating export-competent mRNPs. Overall this is now a very nice piece of work that will be of great interest in the field.

We are grateful for the reviewer's positive feedback.

Reviewer #2:

The authors have addressed essentially all our points satisfactorily. We note, the relevance of the transcript comparisons between the LENG8 and GANP knockdowns would be much clearer if these were done in parallel in the same conditions, or at least had additional controls or explanations demonstrating their equivalence. There seem to be many variables between the two datasets that could contribute to the differences seen – different cell line used, knockdown efficiency, and other potential differences in experimental details, e.g. method and efficiency of nuclear/cytoplasmic fractionation. However, this is a point that another review had a strong focus on, so we're happy to leave this to them and the editor. Otherwise, this is overall a beautiful piece of work.

We thank the reviewer for providing valuable comments. We acknowledge the reviewer's concern regarding the comparison between LENG8 and GANP. In this revised manuscript, we have revised Figure 8 to simply present the RNA features of LENG8-dependent mRNAs.

Reviewer #3:

Overall, Clarke et. al addressed reviewer comments thoroughly and have improved the quality of an already rigorous manuscript. The EMSA experiments with AMPPNP aid in clarifying that ATP hydrolysis is not required for DDX39B unloading and the additional experiments directly comparing ATPase and unloading activities strengthens the conclusions drawn. Edits to text and figures for improved background and clarity based on suggestions are appreciated. The primary issues remaining relate to in cell experiments addressed in original Major Point #3. While RNA sequencing and additional immunofluorescence experiments were performed to address these points, concerns remain regarding implementation and interpretation. These can be addressed by amending figure display and text, not requiring additional experiments. The concerns remaining are:

We appreciate the reviewer's insightful comments and have made revisions accordingly.

1. Direct comparison of “GANP-dependent” and “LENG8-dependent” RNA features in figure 8. The original concern regarding the argument of overlapping vs distinct targets between GANP and LENG8 still stands. While the RNA sequencing with siLENG8 is valuable and provides great insight into changes in gene expression upon LENG8 depletion, direct comparison to previously published siGANP data seems to be inappropriate. This data comes from a different cell line (HCT116 vs A549 here) and as such there likely are differences in these transcriptomes which could influence the impacts of depletions. Furthermore, nuclear vs cytoplasmic isolation was performed differently, and gene expression analysis was performed by microarray which has significantly less depth than RNA sequencing done here. For these reasons, it is suggested that Figure 8 is adapted to not include the “GANP-dependent” plots. If wanting to make a comparison of “LENG8-dependent” features, it would be more appropriate to compare to features of the overall transcriptome in untreated A549 control cells. In order to make a strong point about shared vs distinct targets of GANP and LENG8, depletion and gene expression analyses would need to be done with the same experimental conditions. Text in paragraphs starting at lines 349 and 437 should be amended according to any changes made.

Response: Following the reviewer’s suggestion, rather than emphasizing the difference between LENG8 and GANP, we now simply focus on LENG8-dependent mRNAs in the revised Figure 8. Analysis of shared RNA features of LENG8-dependent mRNAs was performed by comparing them to control genome-wide protein-coding mRNAs. Our results show that LENG8 (TREX-2.1) modulates the nucleocytoplasmic distribution of a subset of mRNAs with high GC content, high exon number, and long transcript length, as compared to the control mRNA population.

2. Evidence of LENG8 function within nuclear speckles is weak.

Immunofluorescence for SC35 to corroborate LENG8 colocalization with nuclear speckles is appreciated. However, overall colocalization is not all that strong and a large majority of LENG8 is localized outside of SC35 signal. Furthermore, there is still no staining for DDX39B. While a role for DDX39B in nuclear speckles is well documented, there is no evidence that DDX39B and LENG8 co-occupy the same speckles and may be residing in distinct pools of speckles.

Response: In this revision, we have removed the immunofluorescence data from Supplementary Fig 1, as it is not the focus of this study. This experiment was conducted during the first revision in response to reviewers’ queries. Further characterization of the detailed cellular context in which LENG8 and DDX39B(UAP56) coordinate nuclear mRNP remodeling will be an interesting direction for future studies.

3. More precise language when describing role of TREX2.1 in nuclear export. Even with added RNA sequencing, experiments still do not strongly support a direct role for TREX2.1 in mRNA export. Authors note “...we selected mRNAs whose total cell levels are not altered but are retained in the nucleus upon LENG8 knockdown. In this manner, we selected the mRNAs whose nucleocytoplasmic transport is affected, excluding other effects on mRNA levels.” However, while this does determine transcripts with impacted N/C ratios in the event of LENG8 knockdown, it does not prove this is due to direct disruption in export, nor exclude the possibility this stems from defects in other processes. For example, decreased nuclear surveillance would also lead to higher N/C ratios, although not a defect in export mechanisms per se. Furthermore, as cited on line 421, LENG8 does associate with quality control related factors, making this a distinct possibility for the changed N/C ratios observed in the sequencing data. With this in mind, the title on line 327 should be edited to read “influences nucleocytoplasmic ratio” rather than

“mediates nuclear export” and edits should be made in this section to not overstate the role, as well as in the discussion starting at line 437. While GANP was demonstrated to play a direct role in export, many of the same experiments leading to this conclusion were not repeated here, and so LENG8 cannot be assumed to play this same role without proper supporting data.

Response: In response to the reviewer’s comments, we have revised the statements regarding the role of LENG8 in mRNA nuclear export. The title of the RNA-seq results has been changed to “LENG8 (TREX-2.1) influences the nucleocytoplasmic ratio of a subset of mRNAs”, as suggested by the reviewer. We have also edited relevant statements throughout the manuscript, including the abstract (line 11-13), introduction (line 89-91), results (line 344-360), and discussion (line 430-444) sections related to the RNA-seq data.